# lpNTK: Better Generalisation with Less Data via Sample Interaction During Learning

**Shangmin Guo**[†] , **Yi Ren**[‡], **Stefano V. Albrecht**[†], **Kenny Smith**[†]
[†] University of Edinburgh, [‡] University of British Columbia,

## Abstract

Although much research has been done on proposing new models or loss functions to improve the generalisation of artificial neural networks (ANNs), less attention has been directed to the impact of the training data on generalisation. In this work, we start from approximating the interaction between samples, i.e. how learning one sample would modify the model's prediction on other samples. Through analysing the terms involved in weight updates in supervised learning, we find that labels influence the interaction between samples. Therefore, we propose the labelled pseudo Neural Tangent Kernel (lpNTK) which takes label information into consideration when measuring the interactions between samples. We first prove that lpNTK asymptotically converges to the empirical neural tangent kernel in terms of the Frobenius norm under certain assumptions. Secondly, we illustrate how lpNTK helps to understand learning phenomena identified in previous work, specifically the learning difficulty of samples and forgetting events during learning. Moreover, we also show that using lpNTK to identify and remove poisoning training samples does not hurt the generalisation performance of ANNs.

## 1 Introduction

In the past decade, artificial neural networks (ANNs) have achieved great successes with the help of large models and very large datasets (Silver et al., 2016; Krizhevsky et al., 2017; Vaswani et al., 2017). There are usually three components involved in training an ANN: the model, i.e. the ANN itself; a loss function; and a dataset, usually labelled for supervised learning. Previous work has shown that generalisation performance can be boosted by changing the model architecture, e.g. ResNet (He et al., 2016) and ViT (Dosovitskiy et al., 2020); or introducing new loss functions, e.g. focal loss (Lin et al., 2017) and regularisation (Goodfellow et al., 2016). Researchers have also explored methods that allow models to choose data, e.g. active learning (Settles, 2012), to improve generalisation performance and reduce laborious data annotation. In this work, we focus on selecting data in order to achieve better generalisation performance.

Although the training samples are usually assumed to be independent and identically distributed (i.i.d, Goodfellow et al., 2016), ANNs do not learn the samples independently, as the parameter update on one labelled sample will influence the predictions for many other samples. This dependency is a double-edged sword. On the plus side, the dependency between seen and unseen samples makes it possible for ANNs to output reliable predictions on held-out data (Koh & Liang, 2017). However, dependencies between samples can also cause catastrophic forgetting (Kirkpatrick et al., 2017), i.e. updates on a sample appearing later might erase the correct predictions for previous samples. Hence, we argue that it is necessary to investigate the relationships between samples in order to understand how data selection affects the learning, and further how to improve generalisation performance through manipulating data.

We first decompose the learning dynamics of ANN models in classification tasks, and show that the interactions between training samples can be well-captured by combining the empirical neural tangent kernel (eNTK Jacot et al., 2018) with label information. We thus propose a scalar sample similarity metric that measures the influence of an update on one sample to the prediction on another, called labelled pseudo NTK (lpNTK), which extends pseudo NTK (Mohamadi & Sutherland, 2022,

---

Correspondence author: `s.guo@ed.ac.uk`

pNTK) by incorporating the label information. As shown in Section 2.3, our lpNTK can be viewed as a linear kernel on a feature space representing each sample as a vector derived from the lpNTK. Since the inner product of high-dimensional vectors can be positive/negative/zero, we point out that there are three types of relationships between a pair of samples: interchangeable, unrelated, and contradictory. Following the analysis of inner products between lpNTK representations, we find that a sample can be considered as redundant if its largest inner product is not with itself, i.e. it can be removed without reducing the trained ANNs' generalisation performance. Moreover, through experiments, we verify that two concepts from previous work can be connected to and explained by the above three types of relationships: the learning difficulty of data discussed by Paul et al. (2021), and the forgetting events during ANN learning found by Toneva et al. (2018). Furthermore, inspired by the discussion about the connections between learning difficulty and generalisation from Sorscher et al. (2022), we show that the generalisation performance of trained ANNs is not influenced, and can be potentially improved, by removing part of the samples in the largest cluster obtained through farthest point clustering (FPC) with lpNTK as the similarity metric.

In summary, we make three contributions in this work: 1) we introduce a new kernel, lpNTK, which can take label information into account for ANNs to measure the interaction between samples during learning; 2) we provide a unified view to explain the learning difficulty of samples and forgetting events using the three types of relationships defined under lpNTK; 3) we show that generalisation performance in classification problems is not impacted by carefully removing data items that have similar lpNTK feature representations.

## 2 LPNTK: SAMPLE INTERACTION VIA FIRST-ORDER TAYLOR APPROXIMATION

### 2.1 DERIVATION ON SUPERVISED LEARNING FOR CLASSIFICATION PROBLEMS

We start from the first-order Taylor approximation to the interactions between two samples in classification. Suppose in a $K$-way classification problem, the dataset $\mathcal{D}$ consists of $N$ labelled samples, i.e. $\mathcal{D} = \{(\boldsymbol{x}_i, y_i)\}_{i=1}^N$. Our neural network model is $f(\boldsymbol{x}; \boldsymbol{w}) \triangleq \boldsymbol{q}(\boldsymbol{x}) = \mathrm{softmax}(\boldsymbol{z}(\boldsymbol{x}; \boldsymbol{w}))$ where $\boldsymbol{w} \in \mathbb{R}^d$ is the vectorised parameters, $\boldsymbol{z} : \mathbb{X} \to \mathbb{R}^K$, and $\boldsymbol{q} \in \Delta^{K-1}$. To update parameters, we assume the cross-entropy loss function, i.e. $L(y, \boldsymbol{q}(\boldsymbol{x})) = -\log \boldsymbol{q}_y(\boldsymbol{x})$ where $\boldsymbol{q}_k$ represents the $k$-th element of the prediction vector $\boldsymbol{q}$, and the back-propagation algorithm (Rumelhart et al., 1986). At time $t$, suppose we take a step of stochastic gradient descent (SGD) with learning rate $\eta$ on a sample $\boldsymbol{x}_u$, we show that this will modify the prediction on $\boldsymbol{x}_o \neq \boldsymbol{x}_u$ as:

$$\Delta_{\boldsymbol{x}_u} \boldsymbol{q}(\boldsymbol{x}_o) \triangleq \boldsymbol{q}^{t+1}(\boldsymbol{x}_o) - \boldsymbol{q}^t(\boldsymbol{x}_o) \approx \eta \cdot \underbrace{\boldsymbol{A}^t(\boldsymbol{x}_o)}_{K \times K} \cdot \underbrace{\boldsymbol{K}^t(\boldsymbol{x}_o, \boldsymbol{x}_u)}_{K \times K} \cdot \underbrace{(\boldsymbol{p}_{\mathrm{tar}}(\boldsymbol{x}_u) - \boldsymbol{q}^t(\boldsymbol{x}_u))}_{K \times 1} \quad (1)$$

where $\boldsymbol{A}^t(\boldsymbol{x}_o) \triangleq \nabla_{\boldsymbol{z}} \boldsymbol{q}^t(\boldsymbol{x}_o)|_{\boldsymbol{z}^t}$, $\boldsymbol{K}^t(\boldsymbol{x}_o, \boldsymbol{x}_u) \triangleq \nabla_{\boldsymbol{w}} \boldsymbol{z}^t(\boldsymbol{x}_o)|_{\boldsymbol{w}^t} \cdot (\nabla_{\boldsymbol{w}} \boldsymbol{z}^t(\boldsymbol{x}_u)|_{\boldsymbol{w}^t})^{\mathsf{T}}$, and $\boldsymbol{p}_{\mathrm{tar}}(\boldsymbol{x}_u)$ is a one-hot vector where only the $y_u$-th element is 1. The full derivation is given in Appendix A.

$\boldsymbol{K}^t(\boldsymbol{x}_o, \boldsymbol{x}_u)$ in Equation 1 is a dot product between the gradient matrix on $\boldsymbol{x}_u$ ($\nabla_{\boldsymbol{w}} \boldsymbol{z}^t(\boldsymbol{x}_u)|_{\boldsymbol{w}^t}$) and $\boldsymbol{x}_o$ ($\nabla_{\boldsymbol{w}} \boldsymbol{z}^t(\boldsymbol{x}_o)|_{\boldsymbol{w}^t}$). Thus, an entry $K_{ij}^t > 0$ means that the two gradient vectors, $\nabla_{\boldsymbol{w};i,.} \boldsymbol{z}^t(\boldsymbol{x}_u)$ and $\nabla_{\boldsymbol{w};j,.} \boldsymbol{z}^t(\boldsymbol{x}_o)$, are pointing in similar directions, thus learning one leads to better prediction on the other; when $K_{ij}^t < 0$ the two gradients are pointing in opposite directions, thus learning one leads to worse prediction on the other. Regarding the case where $K_{ij}^t \approx 0$, the two gradient vectors are approximately orthogonal, and learning one of them has almost no influence on the other. Since $\boldsymbol{K}^t(\boldsymbol{x}_o, \boldsymbol{x}_u)$ is the only term that involves both $\boldsymbol{x}_u$ and $\boldsymbol{x}_o$ at the same time, we argue that the matrix itself or a transformation of it can be seen as a similarity measure between $\boldsymbol{x}_u$ and $\boldsymbol{x}_o$, which naturally follows from the first-order Taylor approximation. $\boldsymbol{K}^t(\boldsymbol{x}_o, \boldsymbol{x}_u)$ is also known as the eNTK (Jacot et al., 2018) in vector output form.

Although $\boldsymbol{K}^t(\boldsymbol{x}_o, \boldsymbol{x}_u)$ is a natural similarity measure between $\boldsymbol{x}_o$ and $\boldsymbol{x}_u$, there are two problems. First, it depends on both the samples and parameters $\boldsymbol{w}^t$, thus becomes variable over training since $\boldsymbol{w}^t$ varies over $t$. Therefore, to make this similarity metric more accurate, it is reasonable to select the trained model which performs the best on the validation set, i.e. parameters having the best generalisation performance. Without loss of generality, we denote such parameters as $\boldsymbol{w}$ in the following, and the matrix as $\boldsymbol{K}$. The second problem is that we can compute $\boldsymbol{K}(\boldsymbol{x}, \boldsymbol{x}')$ for every pair

of samples $(\boldsymbol{x}, \boldsymbol{x}')$, but need to convert them to scalars $\kappa(\boldsymbol{x}, \boldsymbol{x}')$ in order to use them as a conventional scalar-valued kernel. This also makes it possible to directly compare two similarity matrices, e.g. $\boldsymbol{K}$ and $\boldsymbol{K}'$. It is intuitive to use $L_{p,q}$ or Frobenius norm of matrices to convert $\boldsymbol{K}(\boldsymbol{x}, \boldsymbol{x}')$ to scalars. However, through experiments, we found that the off-diagonal entries in $\boldsymbol{K}^t(\boldsymbol{x}_o, \boldsymbol{x}_u)$ are usually non-zero and there are many negative entries, thus neither $L_{p,q}$-norm nor F-norm is appropriate. Instead, the findings from Mohamadi & Sutherland (2022) suggests that the sum of all elements in $\boldsymbol{K}(\boldsymbol{x}, \boldsymbol{x}')$ divided by $K$ serves as a good candidate for this proxy scalar value, and they refer to this quantity as pNTK. We follow their idea, but show that the signs of elements in $\boldsymbol{p}_{\text{tar}}(\boldsymbol{x}) - \boldsymbol{q}(\boldsymbol{x})$ should be considered in order to have higher test accuracy in practice. We provide empirical evidence that this is the case in Section 4.

## 2.2 Information From Labels

Beyond the similarity matrix $\boldsymbol{K}$, there are two other terms in Equation 1, the softmax derivative term $\boldsymbol{A}$ and the prediction error term $\boldsymbol{p}_{\text{tar}}(\boldsymbol{x}_u) - \boldsymbol{q}^t(\boldsymbol{x}_u)$. As shown in Equation 5 from Appendix A, the signs in $\boldsymbol{A}$ are constant across all samples. Thus, it is not necessary to take $\boldsymbol{A}$ into consideration when we measure the similarity between two specific samples.

We now consider the prediction error term. In common practice, $\boldsymbol{p}_{\text{tar}}(\boldsymbol{x})$ is usually a one-hot vector in which only the $y$-th element is 1 and all others are 0. Since $\boldsymbol{q} = \text{softmax}(\boldsymbol{z}) \in \Delta^K$, we can perform the following analysis of the signs of elements in the prediction error term:

$$\boldsymbol{p}_{\text{tar}}(\boldsymbol{x}_u) - \boldsymbol{q}(\boldsymbol{x}_u) = \begin{bmatrix} 0 - q_1 < 0 \\ \dots \\ 1 - q_y > 0 \\ \dots \\ 0 - q_K < 0 \end{bmatrix} \Rightarrow \quad \boldsymbol{s}(y_u) \triangleq \text{sign}\left(\boldsymbol{p}_{\text{tar}}(\boldsymbol{x}_u) - \boldsymbol{q}(\boldsymbol{x}_u)\right) = \begin{bmatrix} -1 \\ \dots \\ +1 \\ \dots \\ -1 \end{bmatrix}. \quad (2)$$

The above analysis shows how learning $\boldsymbol{x}_u$ would modify the predictions on $\boldsymbol{x}_o$, i.e. $\Delta_{\boldsymbol{x}_u} \boldsymbol{q}(\boldsymbol{x}_o)$. Conversely, we can also approximate $\Delta_{\boldsymbol{x}_o} \boldsymbol{q}(\boldsymbol{x}_u)$ in the same way. In this case, it is easy to see that $\boldsymbol{K}(\boldsymbol{x}_u, \boldsymbol{x}_o) = \boldsymbol{K}(\boldsymbol{x}_o, \boldsymbol{x}_u)^{\mathsf{T}}$ and all elements of $\boldsymbol{p}_{\text{tar}}(\boldsymbol{x}_o) - \boldsymbol{q}(\boldsymbol{x}_o)$ are negative except the $y_o$-th digit, since $\boldsymbol{K}(\boldsymbol{x}_u, \boldsymbol{x}_o) \cdot (\boldsymbol{p}_{\text{tar}}(\boldsymbol{x}_o) - \boldsymbol{q}(\boldsymbol{x}_o)) = \boldsymbol{K}(\boldsymbol{x}_o, \boldsymbol{x}_u)^{\mathsf{T}} \cdot (\boldsymbol{p}_{\text{tar}}(\boldsymbol{x}_o) - \boldsymbol{q}(\boldsymbol{x}_o))$. Therefore, for a pair of samples $\boldsymbol{x}_o$ and $\boldsymbol{x}_u$, their labels would change the signs of the rows and columns of the similarity matrix $\boldsymbol{K}(\boldsymbol{x}_o, \boldsymbol{x}_u)$, respectively.

Note that we do not use the whole error term $\boldsymbol{p}_{\text{tar}}(\boldsymbol{x}) - \boldsymbol{q}(\boldsymbol{x})$ but only the signs of its entries. As illustrated in Section 3.1 below, if a large fraction of samples in the dataset share similar gradients, all their prediction errors would become tiny after a relatively short period of training. In this case, taking the magnitude of $\boldsymbol{p}_{\text{tar}}(\boldsymbol{x}) - \boldsymbol{q}(\boldsymbol{x})(\approx \boldsymbol{0})$ into consideration leads to a conclusion that those samples are dissimilar to each other because the model can accurately fit them. Therefore, we argue that the magnitudes of prediction errors are misleading for measuring the similarity between samples from the learning dynamic perspective.

## 2.3 Labelled Pseudo Neural Tangent Kernel (lpNTK)

Following the analysis in Section 2.2, we introduce the following lpNTK:

$$\text{lpNTK}((\boldsymbol{x}_o, y_o), (\boldsymbol{x}_u, y_u)) \triangleq \frac{1}{K} \sum \left[ \boldsymbol{s}(y_u) \cdot \boldsymbol{s}(y_o)^{\mathsf{T}} \right] \odot \boldsymbol{K}(\boldsymbol{x}_o, \boldsymbol{x}_u)$$

$$= \underbrace{\left[ \frac{1}{\sqrt{K}} \boldsymbol{s}(y_o)^{\mathsf{T}} \nabla_{\boldsymbol{w}} \boldsymbol{z}(\boldsymbol{x}_o) \right]}_{1 \times d} \cdot \underbrace{\left[ \nabla_{\boldsymbol{w}} \boldsymbol{z}(\boldsymbol{x}_u)^{\mathsf{T}} \boldsymbol{s}(y_u) \frac{1}{\sqrt{K}} \right]}_{d \times 1} \quad (3)$$

where $\odot$ denotes that element-wise product between two matrices, and $\boldsymbol{s}(\cdot)$ is defined in Equation 2.

The first line of Equation 3 is to emphasise the difference between our lpNTK and pNTK, i.e. we sum up the element-wise products between $\frac{1}{K} \boldsymbol{s}(y_u) \cdot \boldsymbol{s}(y_o)^{\mathsf{T}}$ and $\boldsymbol{K}$. Equivalently, the second line shows that our kernel can also be thought of as a linear kernel on the feature space where a sample $\boldsymbol{x}$ along with its label $y$ is represented as $\frac{1}{\sqrt{K}} \boldsymbol{s}(y)^{\mathsf{T}} \nabla_{\boldsymbol{w}} \boldsymbol{z}(\boldsymbol{x})$. The second expression also shows the novelty of our lpNTK, i.e. the label information is taken into the feature representations of samples. We verify that the gap between lpNTK and eNTK is bounded in Appendix B.

**Practical pipeline**: make our lpNTK more accurately approximate the interactions between samples in practice, we follow the pipeline: 1) fit the model on a given benchmark, and select the parameters $\boldsymbol{w}^*$ which has the best performance on validation set; 2) calculate the lpNTK matrix based on the $\boldsymbol{w}^*$ by enumerating $\kappa((\boldsymbol{x}, y), (\boldsymbol{x}', y'); \boldsymbol{w})$ for all pairs $(\boldsymbol{x}, y)$ and $(\boldsymbol{x}', y')$ from the training set.

## 3    RETHINKING EASY/HARD SAMPLES FOLLOWING LPNTK

Since each labelled sample $(\boldsymbol{x}, y)$ corresponds to a vector $\frac{1}{\sqrt{K}} \boldsymbol{s}(y)^{\mathsf{T}} \nabla_{\boldsymbol{w}} \boldsymbol{z}(\boldsymbol{x}) \in \mathbb{R}^d$ under lpNTK, and the angles between any two such vectors could be acute, right, or obtuse, it is straightforward to see the following three types of relationships between two labelled samples:

• *interchangeable* samples (where the angle is acute) update the parameters of the model in similar directions, thus learning one sample makes the prediction on the other sample also more accurate;

• *unrelated* samples (where the angle is right) update parameters in (almost) orthogonal directions, thus learning one sample (almost) doesn't modify the prediction on the other sample;

• *contradictory* samples (where the angle is obtuse) update parameters in opposite directions, thus learning one sample makes the prediction on the other sample worse.

More details about the above three types of relationships can be found in Appendix C. In the remainder of this section, we illustrate and verify how the above three types of relationships provide a new unified view of some existing learning phenomena, specifically easy/hard samples discussed by Paul et al. (2021) and forgetting events found by Toneva et al. (2018).

### 3.1    A UNIFIED VIEW FOR EASY/HARD SAMPLES AND FORGETTING EVENTS

Suppose at time-step $t$ of training, we use a batch of data $\mathbb{B}^t = \{(\boldsymbol{x}_i^t, y_i^t)\}_{i=1}^{|\mathbb{B}^t|}$ to update the parameters. We now consider how this update would influence the predictions of samples in $\mathbb{B}^{t-1}$ and $\mathbb{B}^{t+1}$. As shown by Equation 1, for a sample $\boldsymbol{x}'$ in either $\mathbb{B}^{t-1}$ or $\mathbb{B}^{t+1}$, the change of its prediction is approximately $\eta \cdot \boldsymbol{A}^t(\boldsymbol{x}') \cdot \sum_{\boldsymbol{x}^t \in \mathbb{B}^t} \boldsymbol{K}^t(\boldsymbol{x}', \boldsymbol{x}^t) \cdot (\boldsymbol{p}_{\text{tar}}(\boldsymbol{x}^t) - \boldsymbol{q}^t(\boldsymbol{x}^t))$. Note that the sum on $\mathbb{B}^t$ is still linear, as all elements in Equation 1 are linear.

Following the above analysis, it is straightforward to see that if a sample in $\mathbb{B}^{t-1}$ is *contradictory* to most of the samples in $\mathbb{B}^t$, it would most likely be *forgotten*, as the updates from $\mathbb{B}^t$ modify its prediction in the "wrong" direction. Furthermore, if the probability of sampling data contradictory to $(\boldsymbol{x}, y)$ is high across the training, then $(\boldsymbol{x}, y)$ would become hard to learn as the updates from this sample are likely to be cancelled out by the updates from the contradictory samples until the prediction error on the contradictory samples are low.

On the other hand, if a sample in $\mathbb{B}^{t+1}$ is *interchangeable* with most of the samples in $\mathbb{B}^t$, it would be an *easy* sample, as the updates from $\mathbb{B}^t$ already modified its prediction in the "correct" direction. Moreover, if there is a large group of samples that are interchangeable with each other, they will be easier to learn since the updates from all of them modify their predictions in similar directions.

Therefore, we argue that the easy/hard samples as well as the forgetting events are closely related to the interactions between the lpNTK feature representations, i.e. $\frac{1}{\sqrt{K}} \boldsymbol{s}(y)^{\mathsf{T}} \nabla_{\boldsymbol{w}} \boldsymbol{z}(\boldsymbol{x})$, of the samples in a dataset. The number of interchangeable/contradictory samples can be thought as the "learning weights" of their corresponding lpNTK feature representations.

An example of and further details about the above analysis are given in Appendix D.1.

### 3.2    EXPERIMENT 1: CONTROL LEARNING DIFFICULTY

The first experiment we run is to verify that the learning difficulty of samples is mainly due to the interactions between them, rather than inherent properties of individual samples. To do so, we empirically verify the following two predictions derived from our explanation given in Section 3.1:

**Prediction 1** for a given universal set of training samples $\mathbb{T}$, for a subset of it $\tilde{\mathbb{T}} \subset \mathbb{T}$ contains fewer samples, the interactions in $\tilde{\mathbb{T}}$ are weaker compared to the interactions in $\mathbb{T}$, thus the learning difficulty of samples in $\tilde{\mathbb{T}}$ is less correlated with their learning difficulty in $\mathbb{T}$;

Figure 1: Correlation between the learning difficulty of samples trained on subsets of varying sizes. For each subfigure, the x-value is the difficulty of samples trained on the universal set (4096), while y-axis is the difficulty trained on the contrast setting (1024, 256, etc.). Smaller Pearson correlation coefficient $\rho$ means less correlation between x and y values. The title of each panel is the settings we compare, e.g. "4096 vs 1" means that we plot the learning difficulty of the same sample on the universal set (of size 4096) against its learnability in a dataset containing just itself.

**Prediction 2** for a given training sample $(\boldsymbol{x}, y)$, if the other samples in the training set $(\boldsymbol{x}', y') \in \mathbb{T} \setminus \{(\boldsymbol{x}, y)\}$ are more interchangeable to it, then $(\boldsymbol{x}, y)$ is easier to learn.

**To verify Prediction 1**, we show that the learning difficulty of a sample on the universal dataset $\mathbb{T}$ becomes less correlated with its learning difficulty on subsets of $\mathbb{T}$ of smaller sizes. Following the definition from Jiang et al. (2021) and Ren et al. (2022), we define the learning difficulty of a sample $(\boldsymbol{x}, y)$ as the integration of its training loss over epochs, or formally $\sum_t l(\boldsymbol{x}, y)$ where $t$ indicates the index of training epochs. In this way, larger values indicate greater learning difficulty.

We run our experiment using the ResNet-18 model (He et al., 2016) and a subset of the CIFAR-10 dataset (Krizhevsky et al., 2009): we select 4096 samples randomly from all the ten classes, giving a set of 40960 samples in total. On this dataset, we first track the learning difficulty of samples through a single training run of the model. Then, we randomly split it into $\frac{4096}{X}$ subsets where $X \in \{1, 4, 16, 256, 1024\}$, and train a model on each of these subsets. For example, when $X = 1$, we train 4096 models of the same architecture and initial parameters on the 4096 subsets containing just 1 sample per class. In all runs, we use the same hyperparameters and train the network with the same batch size. The results are shown in Figure 1. To eliminate the effects from hyperparameters, we also run the same experiment with varying settings of hyperparameters on both CIFAR10 (with ResNet-18, and MNIST where we train a LeNet-5 model (LeCun et al., 1989). Note that we train ResNet-18 on CIFAR10 and LeNet-5 on MNIST in all the following experiments. Further details can be found in Appendix D.2.

Given our analysis in Section 3.1, we expect that the interactions between samples will be more different to the universal set when the dataset size is smaller, as a result the learning difficulty of samples would change more. As shown in Figure 1, the correlation between the learning difficulty on the whole dataset and the subsets indeed becomes less when the subset is of smaller size, which matches with our prediction.

**To verify Prediction 2**, we follow an analysis in Section 3.1, i.e. if there is a large group of samples that are interchangeable/contradictory to a sample, it should become easier/harder to learn respectively. To do so, we first need to group the labelled samples. Considering that the number of sample are usually tens of thousands and the time for computing a $\kappa(\boldsymbol{x}, \boldsymbol{x}')$ is quadratic to the number of samples, we choose the farthest point clustering (FPC) algorithm proposed by Gonzalez (1985), since: 1) in theory, it can be sped up to $\mathcal{O}(N \log M)$ where $M$ is the number of centroids (Feder & Greene, 1988); 2) we cannot arbitrarily interpolate between two gradients[1]. More details are given in Appendix D.5.

As shown in the Appendix D.5, the clusters from FPC are distributed in a heavy long-tail fashion, i.e. most of the samples are in the largest cluster. A straightforward prediction following our method is that the centroid of the head (largest) cluster becomes easier due to the large number of interchangeable samples in the cluster. To verify this, we single out the centroids of all classes on MNIST and CIFAR10, and manually add an equal number of the following three types of samples: 1) most interchangeable samples from the head cluster which should make them easier; 2) most non-interchangeable samples[2] from tail clusters which make them harder; 3) medium interchangeable samples from the "edge" of the head cluster which should have an intermediate effect

---

[1] Clustering algorithms like $k$-means may interpolate between two samples to create centroids of clusters.

[2] We found in our experiment that contradictory samples are rare in practice.

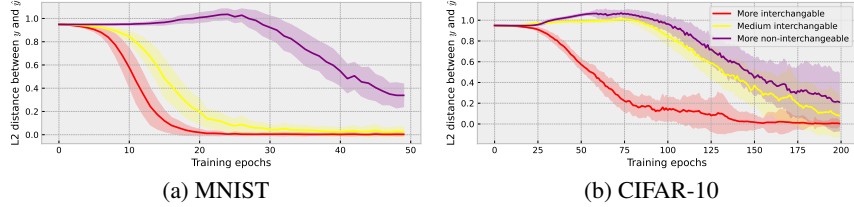

(a) MNIST                    (b) CIFAR-10

Figure 2: Learning difficulty of target samples, i.e. centroids of FPC head clusters, when the samples in the training set are interchangeable (red line), non-interchangeable (purple line), or medium interchangeable (yellow line). The plots show that the learning difficulty of target samples can be controlled through adding samples with different relationships to them into the training sets.

on learnability. We then track the learning difficulty of the centroids on these three types of datasets over 100 random seeds, and the results are shown in Figure 2. It can be seen that the learning difficulty of the target samples become significantly lower when there are more interchangeable samples in the training set, as the red lines are always the lowest. Conversely, as the purple lines are always the highest, the learning difficulty of targets become significantly lower when there exist more non-interchangeable samples in the training set. Similarly, if the samples are medium interchangeable to the centroids, their learning difficulty is just between the other two cases. This shows that we can control the learning difficulty of a sample through controlling the relationships of other samples in the same dataset to it, which supports our Prediction 2 and further our analysis in Section 3.1. More details can be found in Appendix D.3.

## 3.3 EXPERIMENT 2: PREDICT FORGETTING EVENTS WITH lpNTK

The second experiment we run is to check whether the forgetting events can be predicted with lpNTK ($\kappa$). However, since the sum operation in Equation 3 is irreversible, in order to predict the forgetting events with higher accuracy, we propose a variant of lpNTK $\tilde{\kappa}$ by replacing $s(y)$ in the lpNTK representation with $[-\frac{1}{K}, \ldots, 1 - \frac{1}{K}, \ldots, -\frac{1}{K}]^\top$ where $1 - \frac{1}{K}$ is the $y$-th element. More details of this lpNTK variant are given in Appendix D.4. Following Toneva et al. (2018), we say a sample is forgotten at epoch $t+1$ if the model can correctly predict it at $t$ but then makes an incorrect prediction at $t + 1$. To guarantee the accuracy of the first-order Taylor approximation, we set the learning rate to $10^{-3}$ in this experiment. In order to sample enough many forgetting events, we track the predictions on batches of 128 samples over 10 random seeds in the first 50 and 700 iterations of the training process on MNIST and CIFAR10 respectively, and observed 7873 forgetting events on MNIST and 2799 forgetting events on CIFAR10. The results of predicting these forgetting events by the variant of lpNTK are given in Table 1.

As can be seen from Table 1, our variant of lpNTK can predict forgetting events significantly better than random guess during the training of models on both MNIST and CIFAR10. The prediction metrics on CIFAR-10 are higher, possibly because the number of parameters of ResNet-18 is greater than LeNet-5, thus the change of each parameter is smaller during a single iteration, which further leads to a more accurate Taylor approximation.

| Benchmarks | Precision | | Recall | | F1-score | |
|---|---|---|---|---|---|---|
| | Mean | Std | Mean | Std | Mean | Std |
| MNIST | 42.72% | ±6.55% | 59.02% | ±7.49% | 49.54% | ±6.99% |
| CIFAR-10 | 49.47% | ±7.06% | 69.50% | ±7.49% | 57.76% | ±7.36% |

Table 1: Performance of predicting forgetting events with our lpNTK on MNIST and CIFAR-10.

## 4 USE CASE: SUPERVISED LEARNING FOR IMAGE CLASSIFICATION

Inspired by the finding from Sorscher et al. (2022) that selective data pruning can lead to substantially better performance than random selection, we also explore whether pruning samples following the clustering results under lpNTK could help to improve generalisation performance in a typical supervised learning task, image classification. In the experiments below, we use the same models and benchmarks as in Section 3.3.

Following the clustering analysis from Appendix D.5, suppose we compose the lpNTK feature representations of samples from a cluster into a single vector. The size of the cluster can then be thought as the weight for that vector during learning. Given the cluster size distribution is heavily long-tailed, the learning would be biased towards the samples in the head cluster. However, we also know that the samples in the head cluster are more interchangeable with each other, i.e. they update parameters in similar directions. Therefore, we ask the following two questions in Section 4.1 and 4.2.

## 4.1 DO WE NEED ALL THOSE INTERCHANGEABLE SAMPLES FOR GOOD GENERALISATION?

To answer this question, we define a sample as *redundant* under lpNTK if the most interchangeable sample to it is not itself. Formally, for a sample $x$, if there exists another sample $x' \neq x$ such that $\mathrm{lpNTK}((x, y), (x', y')) > \mathrm{lpNTK}((x, y), (x, y))$, then $x$ is considered as a redundant sample.[3] To verify that redundant samples identified in this way are indeed not required for accurate generalisation, we removed them from both MNIST and CIFAR-10 to get de-redundant versions of them under lpNTK. To show the necessity of taking the label information into consideration, we also define redundant samples under pNTK in a similar way, i.e. for a sample $x$, if there exists another sample $x' \neq x$ such that $\mathrm{pNTK}(x, x') > \mathrm{pNTK}(x, x)$, then $x$ is considered as a redundant sample under pNTK. Following this definition, we can also get de-redundant MNIST and CIFAR10 with pNTK. The test accuracy over training epochs on the whole training sets of MNIST and CIFAR-10 along with the de-redundant versions of them are shown in Table 2. To eliminate the effects from hyperparameters, we ran this experiment over 10 different random seeds. Moreover, we did not fine-tune any hyperparameters, to make sure that the comparison between different instances of datasets are fair. As a result, the average test accuracy we obtained is not as high as reported in some other works, e.g. Guo et al. (2022).

| Benchmarks | Full | | De-redundant by pNTK | | De-redundant by lpNTK | |
|---|---|---|---|---|---|---|
| | Mean | Std | Mean | Std | Mean | Std |
| MNIST | 99.31% | 0.03% | 99.27% | 0.05% | 99.30% | 0.03% |
| CIFAR10 | 93.28% | 0.06% | 90.93% | 0.29% | 93.17% | 0.23% |

Table 2: Test accuracy over training epochs of models trained with MNIST and CIFAR-10 as well as the de-redundant versions. The results are averaged across 10 different runs, and the corresponding standard deviation is also given. This table shows that removing the redundant samples defined with our lpNTK leads to a generalisation performance w.o. statistically significant difference to the whole set, while pNTK leads to worse performance.

It can be seen in Table 2 that removing the redundant samples under lpNTK leads to almost the same generalisation performance on both MNIST and CIFAR-10 (converged test accuracy obtained with the whole training sets is not significantly higher than accuracy obtained using the de-redundant version using lpNTK: on MNIST, $t(19) = 0.293, p = 0.774$; on CIFAR10, $t(19) = 1.562, p = 0.153$), whereas the de-redundant versions obtained by pNTK leads to significantly worse generalisation performance (on MNIST, $t(19) = 13.718, p \ll 0.01$; on CIFAR10, $t(19) = 26.252, p \ll 0.01$).

Overall, our results suggest that it is not necessary to train on multiple redundant samples (as identified using lpNTK) in order to have a good generalisation performance; the fact that identifying redundant samples using pNTK *does* lead to a reduction in performance shows that taking the label information into account in evaluating the relationships between samples (as we do in lpNTK) indeed leads to a better similarity measure than pNTK in practice.

## 4.2 WILL THE GENERALISATION PERFORMANCE DECREASE IF THE BIAS IN THE DATA TOWARDS THE NUMEROUS INTERCHANGEABLE SAMPLES IS REMOVED?

To answer this question, we found several relevant clues from existing work. Feldman (2020) demonstrated that memorising the noisy or anomalous labels of the long-tailed samples is necessary in order to achieve close-to-optimal generalisation performance. Paul et al. (2021) and Sorscher et al. (2022) pointed out that keeping the hardest samples can lead to better generalisation performance than keeping only the easy samples. Particularly, the results in Figure 1 of Paul et al. (2021)

---

[3]A visualisation of redundant sample definition is given in Appendix E.1.

| Benchmarks | Full | lpNTK | EL2N | GraNd | Forgot Score | Influence-score |
|---|---|---|---|---|---|---|
| MNIST | 99.31($\pm$0.03)% | 99.37($\pm$0.04)% | 99.33($\pm$0.06)% | 99.28($\pm$0.05)% | 99.26($\pm$0.06)% | 99.27($\pm$0.05)% |
| CIFAR10 | 93.28($\pm$0.06)% | 93.55($\pm$0.12)% | 93.32($\pm$0.07)% | 92.87($\pm$0.13)% | 92.64($\pm$0.22)% | 92.53($\pm$0.18)% |

Table 3: Test accuracy of models trained with full training set and various subsets from MNIST and CIFAR10. The results are averaged across 10 different runs, and the standard deviation are also given. The table shows that randomly removing $10\%$ of the samples in the head cluster leads to slightly better generalisation performance than the original datasets.

show that removing $10\%$ or $20\%$ of the easy samples can lead to better test accuracy than training on the full training set. [4]

As discussed in Section 3, both the long-tail clusters and hard samples can be connected to the three types of relationships under our lpNTK. Specifically, samples in the head cluster are more interchangeable with each other, and thus easier to learn, while the samples in the tail clusters are more unrelated or even contradictory to samples in the head cluster and thus harder to learn. Furthermore, labels of the samples in the tail clusters are more likely to be noisy or anomalous, e.g. the "6" shown in the seventh row in Figure 13 looks more like "4" than a typical "6".

Inspired by the connections between clustering results under lpNTK and the previous works, we: 1) remove the redundant samples from the dataset under lpNTK; 2) cluster the remaining samples with FPC and lpNTK; 3) randomly prune $10\%$ of the samples in the largest cluster; 4) compare the test accuracy of models trained with the original datasets and the pruned versions. This sequence of steps involves removing roughly $20\%$ of the total dataset. The results are given in Table 3.

As we can see in Table 3, the sub-dataset pruned by lpNTK actually lead to slightly higher test accuracy than the full training sets of MNIST and CIFAR-10: a $t$-test comparing converged test accuracy after training on the full vs lpNTK-pruned datasets shows significantly higher performance in the pruned sets (MNIST: $t(19) = -3.205, p = 0.005$; CIFAR: $t(19) = -3.996, p = 0.003$). This suggests that it is potentially possible to improve the generalisation performance on test sets by removing some interchangeable samples in the head cluster from FPC on lpNTK.

### 4.3 Limitation of This Initial Exploration of lpNTK

Although the results illustrated above answer the two questions in Section 4.1 and 4.2, this work still suffers from several limitations when applying lpNTK in practice.

*Additional hyper-parameter selection:* our experiments in this section involve decision on several additional hyper-parameters. The fraction of samples to be removed in Section 4.2, $20\%$, is purely heuristic. Similarly, the number of clusters in FPC clustering $M$ is also heuristic. Here, the optimal choice of $M$ is related to both the training and test datasets and needs to be further studied.

*Computational complexity of lpNTK* $\mathcal{O}(N^2 d^2)$ is also a significant limitation for its application in practice. To overcome this, a proxy quantity for lpNTK is necessary, especially a proxy that can be computed in $\mathcal{O}(1)$ along with the training of the models.

*Dependency on accurate models parameters* $\boldsymbol{w}^*$: as stated in Section 2.3, an accurate lpNTK depends on parameters that perform accurate prediction on validation data. This limits the calculation of lpNTK to be post-training. In future work, we aim to provide a method that can compute lpNTK along with the training of models, and improve the final generalisation of models by doing so.

*Relationships between only training samples*: technically, lpNTK can measure any pair of annotated samples. Thus, it is also possible to use lpNTK to measure the similarity between the training and test data, which is very relevant to methods like influence function (Koh & Liang, 2017).

## 5 Related Works

**Transmission Bottleneck in Iterated Learning:** This work originated as an exploration of the possible forms of transmission bottleneck in the iterated learning framework (Kirby & Hurford, 2002; Smith et al., 2003; Kirby et al., 2014) on deep learning models. In existing works (e.g.

---

[4]In the paper (Paul et al., 2021), the authors pruned the samples with lowest EL2N scores to obtain better generalisation performance, which corresponds to the samples in the head cluster discussed in Appendix D.5.

Guo et al., 2020; Lu et al., 2020; Ren et al., 2020; Vani et al., 2021; Rajeswar et al., 2022), the transmission bottleneck is usually implemented as a limit on the number of pre-training epochs for the new generation of agents. However, as shown in human experiments by Kirby et al. (2008), the transmission bottleneck can also be a subset of the whole training set, which works well for human agents. So, inspired by the original work on humans, we explore the possibility of using subset-sampling as a form of transmission bottleneck. As shown in Section 4.2, it is indeed possible to have higher generalisation performance through limiting the size of subsets. Thus, we argue that this work sheds light on a new form of transmission bottleneck of iterated learning for DL agents.

**Neural Tangent Kernel:** The NTK was first proposed by Jacot et al. (2018), and was derived on fully connected neural networks, or equivalently multi-layer perceptrons. Lee et al. (2019); Yang (2019; 2020); Yang & Littwin (2021) then extend NTK to most of the popular neural network architectures. Beyond the theoretical insights, NTK has also been applied to various kinds of DL tasks, e.g. 1) Park et al. (2020) and Chen et al. (2021) in neural architecture search; 2) Zhou et al. (2021) in meta learning; and 3) Holzmüller et al. (2022) and Wang et al. (2021) in active learning. NTK has also been applied in dataset distillation (Nguyen et al., 2020; 2021), which is closely related to our work, thus we discuss it separately in the next paragraph. Mohamadi & Sutherland (2022) explore how to convert the matrix-valued eNTK for classification problems to scalar values. They show that the sum of the eNTK matrix would asymptotically converge to the eNTK, which inspires lpNTK. However, we emphasise the information from labels, and focus more on practical use cases of lpNTK. We also connect lpNTK with the practical learning phenomena like learning difficulty of samples and forgetting events.

**Coreset Selection and Dataset Distillation:** As discussed in Section 4, we improve the generalisation performance through removing part of the samples in the training sets. This technique is also a common practice in coreset selections (CS, Guo et al., 2022), although the aim of CS is usually to select a subset of training samples that can obtain generalisation performance *similar to* the whole set. On the other hand, we aim to show that it is possible to *improve* the generalisation performance via removing samples. In the meantime, as shown in Equation 3, our lpNTK is defined on the gradients of the *outputs* w.r.t the parameters. Work on both coreset selection (Killamsetty et al., 2021) and dataset distillation (Zhao et al., 2021) have also explored the information from gradients for either selecting or synthesising samples. However, these works aimed to match the gradients of the *loss function* w.r.t the parameter on the selected/synthesised samples with the *whole dataset*, whereas we focus on the gradients *between samples* in the training set.

## 6 CONCLUSION

In this work, we studied the impact of data relationships on generalisation by approximating the interactions between labelled samples, i.e. how learning one sample modifies the prediction on the other, via first-order Taylor approximation. With SGD as the optimiser and cross entropy as the loss function, we analysed Equation 1, showed that eNTK matrix is a natural similarity measure and how labels change the signs of its elements. Taking the label information into consideration, we proposed lpNTK in Section 2, and proved that it asymptotically converges to the eNTK under certain assumptions. As illustrated in Section 3, it is then straightforward to see that samples in a dataset might be interchangeable, unrelated, or contradictory. Through experiments on MNIST and CIFAR-10, we showed that the learning difficulty of samples as well as forgetting events can be well explained under a unified view following these three types of relationships. Moreover, we clustered the samples based on lpNTK, and found that the distributions over clusters are extremely long-tailed, which can further support our explanation about the learning difficulty of samples in practice.

Inspired by Paul et al. (2021) and Sorscher et al. (2022), we showed that the generalisation performance does not decrease on both MNIST and CIFAR-10 through pruning out part of the interchangeable samples in the largest cluster obtained via FPC and lpNTK. Our findings also agree with Sorscher et al. (2022) in that the minority of the training samples are important for good generalisation performance, when a large fraction of datasets can be used to train models. Or equivalently, the bias towards the majority samples (those in the largest cluster) may degrade generalisation in such cases. Overall, we believe that our work provides a novel perspective to understand and analyse the learning of DL models through the lens of learning dynamics, label information, and sample relationships.

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

# A  FULL DERIVATION OF LPNTK ON CLASSIFICATION PROBLEM IN SUPERVISED LEARNING

Following the problem formulated in Section 2, let's begin with a Taylor expansion of $\Delta_{\boldsymbol{x}_u} \boldsymbol{q}(\boldsymbol{x}_o)$,

$$\underbrace{\boldsymbol{q}^{t+1}(\boldsymbol{x}_o)}_{K \times 1} - \underbrace{\boldsymbol{q}^t(\boldsymbol{x}_o)}_{K \times 1} = \underbrace{\nabla_{\boldsymbol{w}} \boldsymbol{q}^t(\boldsymbol{x}_o)|_{\boldsymbol{w}^t}}_{K \times d} \cdot \underbrace{(\boldsymbol{w}^{t+1} - \boldsymbol{w}^t)}_{d \times 1} + \mathcal{O}(\|\boldsymbol{w}^{t+1} - \boldsymbol{w}^t\|^2).$$

To evaluate the leading term, we can plug in the definition of SGD and apply the chain rule repeatedly:

$$\underbrace{\nabla_{\boldsymbol{w}} \boldsymbol{q}^t(\boldsymbol{x}_o)|_{\boldsymbol{w}^t}}_{K \times d} \cdot \underbrace{(\boldsymbol{w}^{t+1} - \boldsymbol{w}^t)}_{d \times 1} = \left( \underbrace{\nabla_{\boldsymbol{z}} \boldsymbol{q}^t(\boldsymbol{x}_o)|_{\boldsymbol{z}^t}}_{K \times K} \cdot \underbrace{\nabla_{\boldsymbol{w}} \boldsymbol{z}^t(\boldsymbol{x}_o)|_{\boldsymbol{w}^t}}_{K \times d} \right) \cdot \left( - \eta \underbrace{\nabla_{\boldsymbol{w}} L(\boldsymbol{x}_u)|_{\boldsymbol{w}^t}}_{1 \times d} \right)^{\mathsf{T}}$$

$$= \nabla_{\boldsymbol{z}} \boldsymbol{q}^t(\boldsymbol{x}_o)|_{\boldsymbol{z}^t} \cdot \nabla_{\boldsymbol{w}} \boldsymbol{z}^t(\boldsymbol{x}_o)|_{\boldsymbol{w}^t} \cdot \left( - \eta \nabla_{\boldsymbol{z}} L(\boldsymbol{x}_u)|_{\boldsymbol{z}^t} \cdot \nabla_{\boldsymbol{w}} \boldsymbol{z}^t(\boldsymbol{x}_u)|_{\boldsymbol{w}^t} \right)^{\mathsf{T}}$$

$$= -\eta \underbrace{\nabla_{\boldsymbol{z}} \boldsymbol{q}^t(\boldsymbol{x}_o)|_{\boldsymbol{z}^t}}_{K \times K} \cdot \left[ \underbrace{\nabla_{\boldsymbol{w}} \boldsymbol{z}^t(\boldsymbol{x}_o)|_{\boldsymbol{w}^t}}_{K \times d} \cdot \underbrace{\left( \nabla_{\boldsymbol{w}} \boldsymbol{z}^t(\boldsymbol{x}_u)|_{\boldsymbol{w}^t} \right)^{\mathsf{T}}}_{d \times K} \right] \cdot \underbrace{\left( \nabla_{\boldsymbol{z}} L(\boldsymbol{x}_u)|_{\boldsymbol{z}^t} \right)^{\mathsf{T}}}_{K \times 1}$$

$$= \eta \cdot \boldsymbol{A}^t(\boldsymbol{x}_o) \cdot \boldsymbol{K}^t(\boldsymbol{x}_o, \boldsymbol{x}_u) \cdot \left( \boldsymbol{p}_{\text{tar}}(\boldsymbol{x}_u) - \boldsymbol{q}^t(\boldsymbol{x}_u) \right). \tag{4}$$

For the higher-order terms, using as above that

$$\boldsymbol{w}^{t+1} - \boldsymbol{w}^t = \eta \nabla_{\boldsymbol{w}} \boldsymbol{z}^t(\boldsymbol{x}_u)|_{\boldsymbol{w}^t}^{\mathsf{T}} \cdot \left( \boldsymbol{p}_{\text{tar}}(\boldsymbol{x}_u) - \boldsymbol{q}^t(\boldsymbol{x}_u) \right)$$

and note that $\|\boldsymbol{p}_{\text{tar}}(\boldsymbol{x}_u) - \boldsymbol{q}^t(\boldsymbol{x}_u)\|$ is bounded as $\boldsymbol{p}_{\text{tar}}, \boldsymbol{q} \in \Delta^K$, we have that

$$\mathcal{O}(\|\boldsymbol{w}^{t+1} - \boldsymbol{w}^t\|^2) = \mathcal{O}(\eta^2 \|(\nabla_{\boldsymbol{w}} \boldsymbol{z}(\boldsymbol{x}_u)|_{\boldsymbol{w}^t})^{\mathsf{T}}\|_{\text{op}}^2 \|\boldsymbol{p}_{\text{tar}}(\boldsymbol{x}_u) - \boldsymbol{q}^t(\boldsymbol{x}_u)\|^2) = \mathcal{O}(\eta^2 \|\nabla_{\boldsymbol{w}} \boldsymbol{z}(\boldsymbol{x}_u)\|_{\text{op}}^2). \quad \square$$

The first term in the above decomposition is a symmetric positive semi-definite (PSD) matrix shown below,

$$\boldsymbol{A}^t(\boldsymbol{x}_o) = -\nabla_{\boldsymbol{z}} \boldsymbol{q}^t(\boldsymbol{x}_o)|_{\boldsymbol{z}^t} = \begin{bmatrix} q_1(1 - q_1) & -q_1 q_2 & \cdots & -q_1 q_K \\ -q_2 q_1 & q_2(1 - q_2) & \cdots & -q_2 q_K \\ \vdots & \vdots & \ddots & \vdots \\ -q_K q_1 & -q_K q_2 & \cdots & q_K(1 - q_K) \end{bmatrix}. \tag{5}$$

The second term in the above decomposition, $\boldsymbol{K}^t(\boldsymbol{x}_o, \boldsymbol{x}_u)$, is the outer product of the Jacobian on $\boldsymbol{x}_o$ and $\boldsymbol{x}_u$. It is intuitive to see that the elements in this matrix would be large if the two Jacobians are both large and in similar direction. Since $\boldsymbol{K}^t(\boldsymbol{x}_o, \boldsymbol{x}_u)$ is a matrix rather than a scalar, so it is not a conventional scalar-valued kernel. This matrix is also usually known as matrix-valued NTK.

The third term in the above decomposition is the prediction error on $\boldsymbol{x}_u$, $\boldsymbol{p}_{\text{tar}}(\boldsymbol{x}_u) - \boldsymbol{q}^t(\boldsymbol{x}_u)$. During the training, for most of the samples, this term would gradually approach $\boldsymbol{0}$, suppose the model can fit well on the dataset. However, it is varying over the epochs, so it is not a constant like the sign vector $\boldsymbol{s}(y_u)$. Thus, we introduce only the signs from this vector into our lpNTK, as illustrated in Section 2.

# B  FURTHER RESULTS ON THE ASYMPTOTIC CONVERGENCE OF LPNTK

**Theorem 1.** *(Informal). Suppose the last layer of our neural network $\boldsymbol{z}(\boldsymbol{x}; \boldsymbol{w})$ is a linear layer of width $w$. Let $\kappa((\boldsymbol{x}, y), (\boldsymbol{x}', y'))$ be the corresponding lpNTK, and $\tilde{\boldsymbol{K}}(\boldsymbol{x}, \boldsymbol{x}')$ be the corresponding eNTK. Over the initialisation of $\boldsymbol{z}(\boldsymbol{x}; \boldsymbol{w})$, with high probability, the following holds:*

$$\|\kappa((\boldsymbol{x}, y), (\boldsymbol{x}', y')) \otimes \boldsymbol{I}_K - \tilde{\boldsymbol{K}}(\boldsymbol{x}, \boldsymbol{x}')\|_F \in \mathcal{O}(1) \tag{6}$$

where $\boldsymbol{I}_K$ is a $K$-by-$K$ identity matrix. A formal statement and proof are given in Appendix B.1. We also provide our interpretation of this result in Appendix B.2.

## B.1 PROOF OF THEOREM 1 IN SECTION 2.3

Out proof blow relies on *sub-exponential* random variables, and we recommend the Appendix B.1 of Mohamadi & Sutherland (2022) and notes written by Xiong (2022) for the necessary background knowledge.

Following the notation from Section 2.1, we assume our ANN model $z(\cdot)$ has $L$ layers, and the vectorised parameters of a layer $l \in \{1, 2, \ldots, L\}$ is denoted as $w^l$ whose width is $w_l$. In the meantime, we assume the last layer of $z(\cdot)$ is a dense layer parameterised by a matrix $W^L$, and the input to the last layer is $g(x; w^{\{1,;L-1\}}) \in \mathbb{R}^{w_L}$, thus $z(x) = W^L \cdot g(x)$. Lastly, let's assume the parameters of the last layer are initialised by the LeCun initialisation proposed by LeCun et al. (1998), i.e. $W_{i,j}^L \sim \mathcal{N}(0, \frac{1}{w_L})$. Equivalently, $W_{i,j}^L$ can be seen as $W_{i,j}^L = \frac{1}{\sqrt{w_L}} u$ where $u \sim \mathcal{N}(0, 1)$.

It has been shown by Lee et al. (2019) and Yang (2020) that the eNTK can be formulated as the sum of gradients of the outputs w.r.t the vectorised parameters $w^l$. That said, the eNTK of an ANN $z$ can be denoted as

$$K^z(x, x') = \sum_{l=1}^{L} \nabla_{w^l} z(x) \nabla_{w^l} z(x')^\top. \tag{7}$$

Since the last layer of $z(\cdot)$ is a dense layer, i.e. $z(x) = W^L \cdot g(x)$, we know that $\nabla_{w^l} z(x) = \nabla_g z(x) \cdot \nabla_{w^l} g(x) = w^L \cdot \nabla_{w^l} g(x)$. Thus, the above Equation 7 can be re-written as

$$
\begin{aligned}
K^z(x, x') &= \nabla_{w^L} z(x) \nabla_{w^L} z(x')^\top + \sum_{l=1}^{L-1} w^L \nabla_{w^l} g(x) \nabla_{w^l} g(x')^\top w^{L^\top} \\
&= g(x)^\top g(x') I_K + w^L \left( \sum_{l=1}^{L-1} \nabla_{w^l} g(x) \nabla_{w^l} g(x')^\top \right) w^{L^\top} \\
&= g(x)^\top g(x') I_K + w^L K^g(x, x') w^{L^\top}
\end{aligned}
\tag{8}
$$

where $I_K$ is a $K$-by-$K$ identity matrix.

Then, by expanding Equation 8, we can have that the elements in $K^z(x, x')$ are

$$
\begin{aligned}
K_{ij}^z(x, x') &= \mathbf{1}(i = j) g(x)^\top g(x') + W_{i\cdot}^L K^g(x, x') W_{j\cdot}^{L^\top} \\
&= \mathbf{1}(i = j) g(x)^\top g(x') + \sum_{a=1}^{w_L} \sum_{b=1}^{w_L} W_{ia}^L W_{jb}^L K_{ab}^g(x, x') \\
&= \mathbf{1}(i = j) g(x)^\top g(x') + \frac{1}{w_L} \sum_{a=1}^{w_L} \sum_{b=1}^{w_L} u_{ia} u_{jb} K_{ab}^g(x, x')
\end{aligned}
\tag{9}
$$

where $\mathbf{1}(\cdot)$ is the indicator function.

From Equation 8, like the pNTK proposed by Mohamadi & Sutherland (2022), it can be seen that lpNTK simply calculates a weighted summation of the element in eNTK. Thus, lpNTK can also be seen as appending another dense layer parameterised by $\frac{1}{\sqrt{K}} s(y)$ to the ANN $z(\cdot)$ where $s(y)$ is defined in Equation 2. Similar to how we get Equation 9, we can get that the value of lpNTK between $x$ and $x'$ is

$$
\begin{aligned}
\kappa^z(x, x') &= g(x)^\top g(x') + \frac{1}{K} \sum_{c=1}^{K} \sum_{d=1}^{K} s_c(y) s_d(y') K_{cd}^z(x, x') \\
&= g(x)^\top g(x') + \frac{1}{K w_L} \sum_{c=1}^{K} \sum_{d=1}^{K} \sum_{a=1}^{w_L} \sum_{b=1}^{w_L} s_c(y) s_d(y') u_{ca} u_{db} K_{cd}^g(x, x').
\end{aligned}
\tag{10}
$$

Now, let's define the difference matrix between lpNTK and eNTK as $\boldsymbol{D}(\boldsymbol{x}, \boldsymbol{x}') \triangleq \frac{1}{\sqrt{w_L}}\kappa^{\boldsymbol{z}}(\boldsymbol{x}, \boldsymbol{x}') \bigotimes \boldsymbol{I}_K - \frac{1}{\sqrt{w_L}}\boldsymbol{K_z}(\boldsymbol{x}, \boldsymbol{x}')$, then the entries in $\boldsymbol{D}(\boldsymbol{x}, \boldsymbol{x}')$ are

$$
D_{ij}(\boldsymbol{x}, \boldsymbol{x}') = \frac{1}{w_L\sqrt{w_L}}
\begin{cases}
\sum_{a=1}^{w_L}\sum_{b=1}^{w_L} u_{ia}u_{jb}K_{ab}^{\boldsymbol{g}}(\boldsymbol{x}, \boldsymbol{x}') \\
-\frac{1}{K}\sum_{c=1}^{K}\sum_{d=1}^{K}\sum_{a=1}^{w_L}\sum_{b=1}^{w_L} s_c(y)s_d(y')u_{ca}u_{db}fK_{ab}^{\boldsymbol{g}}(\boldsymbol{x}, \boldsymbol{x}'), & \text{if } i = j \\
\sum_{a=1}^{w_L}\sum_{b=1}^{w_L} u_{ia}u_{jb}K_{ab}^{\boldsymbol{g}}(\boldsymbol{x}, \boldsymbol{x}'), & \text{if } i \neq j.
\end{cases}
\tag{11}
$$

Since all parameters are i.i.d, we can further simplify the above equation as:

$$
D_{ij}(\boldsymbol{x}, \boldsymbol{x}') = \frac{1}{w_L\sqrt{w_L}}\sum_{a=1}^{w_L}\sum_{b=1}^{w_L} K_{ab}^{\boldsymbol{g}}(\boldsymbol{x}, \boldsymbol{x}')
\begin{cases}
u_{ia}u_{jb} - \frac{1}{K}\sum_{c=1}^{K}\sum_{d=1}^{K} s_c(y)s_d(y')u_{ca}u_{db}, & \text{if } i = j \\
u_{ia}u_{jb}, & \text{if } i \neq j.
\end{cases}
\tag{12}
$$

It is then straightforward to see that we can use inequalities for sub-exponential random variables to bound the above result, since: 1) $u_i \sim \mathcal{N}(0, 1)$; 2) $u_i^2 \sim \mathcal{X}^2(1)$ and is sub-exponential with parameters $\nu^2 = 4$ and $\beta = 4$ ($SE(4, 4)$); 3) $u_iu_j \in SE(2, \sqrt{2})$; 4) linear combination of sub-exponential random variables is still sub-exponential; 5) for a random variable $X \in SE(\nu^2, \beta)$ the following inequality holds with probability at least $1 - \delta$:

$$
|X - \mu| < \max\left(\nu^2\sqrt{2\log\frac{2}{\delta}}, 2\beta\log\frac{2}{\delta}\right).
$$

However, note that not all elements in Equation 11 are independent of each other in the case $i = j$. Suppose that $\alpha = ||\boldsymbol{K^g}(\boldsymbol{x}, \boldsymbol{x}')||_\infty$, we can then rewrite the case as:

$$
\begin{aligned}
D_{ii}(\boldsymbol{x}, \boldsymbol{x}') \leq \frac{\alpha}{w_L\sqrt{w_L}}\Bigg[ & \underbrace{\sum_{a=1}^{w_L}\left(u_{ia}^2 - \frac{1}{K}\sum_{c=1}^{K} s_c(y)s_c(y')u_{ca^2}\right)}_{D_{ii}^1(\boldsymbol{x}, \boldsymbol{x}')} \\
& - \underbrace{\frac{1}{K}\sum_{a=1}^{w_L}\sum_{c=1}^{K}\sum_{d=1, d\neq c}^{K} s_c(y)s_d(y')u_{ca}u_{da}}_{D_{ii}^2(\boldsymbol{x}, \boldsymbol{x}')} \\
& + \underbrace{\sum_{a=1}^{w_L}\sum_{b=1, b\neq a}^{w_L} u_{ia}u_{ib}}_{D_{ii}^3(\boldsymbol{x}, \boldsymbol{x}')} \\
& - \underbrace{\frac{1}{K}\sum_{a=1}^{w_L}\sum_{b=1, b\neq a}^{w_L}\sum_{c=1}^{K}\sum_{d=1}^{K} s_c(y)s_d(y')u_{ca}u_{db}}_{D_{ii}^4(\boldsymbol{x}, \boldsymbol{x}')}\Bigg].
\end{aligned}
\tag{13}
$$

In the following, we will bound the above four $D_{ii}(\boldsymbol{x}, \boldsymbol{x}')$ terms separately. Let's start with $D_{ii}^1(\boldsymbol{x}, \boldsymbol{x}')$. Considering that a linear combination of a sequence of sub-exponential random variables $\{U_i\}$ where $U_i \in SE(\nu_i^2, \beta_i)$ is sub-exponential, i.e. $\sum_i \alpha_i U_i \in SE(\sum_i \alpha_i^2\nu_i^2, \max_i |\alpha_i|\beta_i)$,

we can have the following derivation:

$$
\begin{aligned}
D_{ii}^1(\boldsymbol{x}, \boldsymbol{x}') &\leq \frac{\alpha}{w_L\sqrt{w_L}} \sum_{a=1}^{w_L} \left( u_{ia}^2 - \frac{1}{K}\sum_{c=1}^{K} s_c(y)s_c(y')u_{ca^2} \right) \\
&= \frac{\alpha}{w_L\sqrt{w_L}} \sum_{a=1}^{w_L} \left( u_{ia}^2 - \frac{1}{K}s_i(y)s_i(y')u_{ia}^2 - \frac{1}{K}\sum_{c=1,c\neq i}^{K} s_c(y)s_c(y')u_{ca^2} \right) \\
&\in \frac{\alpha}{w_L\sqrt{w_L}} \sum_{a=1}^{w_L} \left( SE\left( 4\cdot \left[\frac{K-s_i(y)s_i(y')}{K}\right]^2, 4\cdot \left[\frac{K-s_i(y)s_i(y')}{K}\right] \right), \right.\\
&\qquad\qquad\qquad \left. -SE\left( 4\cdot\frac{K-1}{K^2}, 4\cdot\frac{1}{K} \right) \right) \\
&= \frac{\alpha}{w_L\sqrt{w_L}} \sum_{a=1}^{w_L} SE\left( 4\cdot\frac{(K-s_i(y)s_i(y'))^2+K-1}{K^2}, 4\cdot\max\left(\frac{K-s_i(y)s_i(y')}{K}, \frac{1}{K}\right) \right) \\
&= \frac{2\alpha}{w_L} \qquad\qquad SE\left( \frac{K^2-(2s_i(y)s_i(y')-1)K}{K^2}, \frac{K-s_i(y)s_i(y')}{\sqrt{w_L}K} \right) \\
&\in \frac{2\alpha}{w_L} \qquad\qquad SE\left( \frac{K+3}{K}, \frac{K+1}{\sqrt{w_L}K} \right)
\end{aligned}
\tag{14}
$$

With $w_L \gg K > 2$ and the property (5) from the paragraph below Equation 11, we can claim:

$$
|D_{ii}^1(\boldsymbol{x},\boldsymbol{x}')| < \frac{2\alpha}{w_L}\frac{(K+3)}{K}\sqrt{2\log\frac{8}{\delta}} \leq \frac{2\alpha}{w_L}2\sqrt{2\log\frac{8}{\delta}}
\tag{15}
$$

with probability at least $1-\frac{\delta}{4}$.

Regarding the remaining terms, the entries are products of two *independent* Gaussian r.v. $u \cdot u'$, thus the weighted sum of them are also sub-exponential. Further, we can bound all of them with the property we used in Equation 15. So, similar to the above analysis of $D_{ii}^1(\boldsymbol{x},\boldsymbol{x}')$, we can claim that:

$$
|D_{ii}^2(\boldsymbol{x},\boldsymbol{x}')| < \frac{2\alpha}{w_L}\max\left( \frac{K-1}{K}\sqrt{2\log\frac{8}{\delta}}, \frac{1}{K}\sqrt{\frac{2}{w_L}}\log\frac{8}{\delta} \right)
\tag{16}
$$

$$
|D_{ii}^3(\boldsymbol{x},\boldsymbol{x}')| < \frac{2\alpha}{\sqrt{w_L}}\max\left( \frac{w_L-1}{w_L}\sqrt{2\log\frac{8}{\delta}}, \frac{\sqrt{2}}{w_L}\log\frac{8}{\delta} \right)
\tag{17}
$$

$$
|D_{ii}^4(\boldsymbol{x},\boldsymbol{x}')| < \frac{2\alpha}{\sqrt{w_L}}\max\left( \frac{w_L-1}{w_L}\sqrt{2\log\frac{8}{\delta}}, \frac{\sqrt{2}}{w_L}\log\frac{8}{\delta} \right)
\tag{18}
$$

with probability at least $1-\frac{\delta}{4}$.

Regarding the case $i \neq j$ of Equation 11, the result is identical to $|D_{ii}^3(\boldsymbol{x},\boldsymbol{x}')|$ shown in the above Equation 17. Therefore, considering Equation 15 to 18 all together, we can loosen the above bounds, and have the following inequalities:

$$
|D_{ij}(\boldsymbol{x},\boldsymbol{x}')| < \frac{8\alpha}{\sqrt{w_L}}\max\left( 2\sqrt{2\log\frac{8}{\delta}}, \sqrt{2}\log\frac{8}{\delta} \right)
\tag{19}
$$

with probability at least $1-\delta$.

Therefore, to sum up from the above, since the Frobenius norm of $\boldsymbol{D}(\boldsymbol{x},\boldsymbol{x}')$ is $\sqrt{\sum_{i,j}D_{i,j}(\boldsymbol{x},\boldsymbol{x}')}$, we can get the following bound for the F-norm of the difference matrix:

$$
Pr\left( ||\boldsymbol{D}(\boldsymbol{x},\boldsymbol{x}')||_F \leq \frac{8K\alpha}{\sqrt{w_L}}\max\left( 2\sqrt{2\log\frac{8K^2}{\delta}}, \sqrt{2}\log\frac{8K^2}{\delta} \right) \right) \geq 1-\delta.
\tag{20}
$$

$\square$

## B.2 Interpretation of Theorem 1

The above proof and Equation 6 basically say that, even under the infinity width assumption, the gap between lpNTK and eNTK is relatively bounded and non-negligible. They are relatively bounded means that lpNTK won't diverge from eNTK to an arbitrarily large value. The gap between lpNTK and eNTK is also non-negligible because the difference doesn't converge to 0 either.

## C Further Details about the Sample Relationships under lpNTK

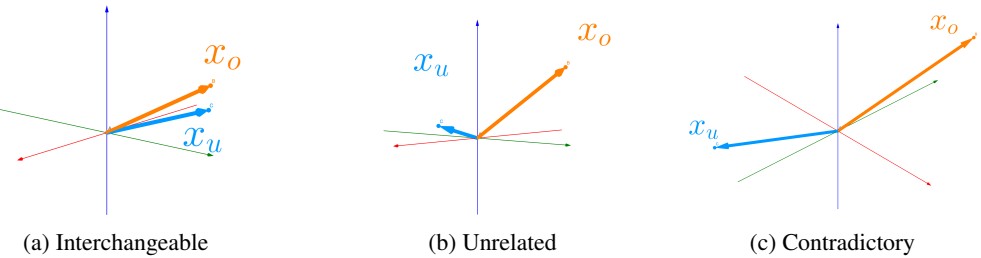

(a) Interchangeable        (b) Unrelated        (c) Contradictory

Figure 3: Visualisation of three possible relationships between data samples. Each vector represents an update to parameters of models based on a back-propagation from a sample e.g. $\boldsymbol{x}_o$ or $\boldsymbol{x}_u$ and their corresponding labels. Formal definitions are given below.

## C.1 Interchangeable

In this case, the two gradient vectors construct a small acute angle, and projections on each other is longer than a fraction $t$ of the other vector, formally the conditions can be written as

$$\frac{\nabla_{\boldsymbol{w}}\boldsymbol{z}_y(\boldsymbol{x}_u) \cdot \nabla_{\boldsymbol{w}}\boldsymbol{z}_y(\boldsymbol{x}_o)}{|\nabla_{\boldsymbol{w}}\boldsymbol{z}_y(\boldsymbol{x}_u)|^2} \geq t \wedge \frac{\nabla_{\boldsymbol{w}}\boldsymbol{z}_y(\boldsymbol{x}_u) \cdot \nabla_{\boldsymbol{w}}\boldsymbol{z}_y(\boldsymbol{x}_o)}{|\nabla_{\boldsymbol{w}}\boldsymbol{z}_y(\boldsymbol{x}_o)|^2} \geq t$$

where $t \in (\frac{1}{2}, 1]$. This situation is visualised in Figure 3a.

## C.2 Unrelated

In this case, the two gradient vectors are almost inter-orthogonal, formally the conditions can be written as

$$\frac{\nabla_{\boldsymbol{w}}\boldsymbol{z}_y(\boldsymbol{x}_u) \cdot \nabla_{\boldsymbol{w}}\boldsymbol{z}_y(\boldsymbol{x}_o)}{|\nabla_{\boldsymbol{w}}\boldsymbol{z}_y(\boldsymbol{x}_u)|^2} \approx 0 \wedge \frac{\nabla_{\boldsymbol{w}}\boldsymbol{z}_y(\boldsymbol{x}_u) \cdot \nabla_{\boldsymbol{w}}\boldsymbol{z}_y(\boldsymbol{x}_o)}{|\nabla_{\boldsymbol{w}}\boldsymbol{z}_y(\boldsymbol{x}_o)|^2} \approx 0.$$

This situation is visualised in Figure 3b.

## C.3 Contradictory

In this case, the two gradient vectors are almost opposite. Like the interchangeable case, the conditions can be formally written as:

$$\frac{\nabla_{\boldsymbol{w}}\boldsymbol{z}_y(\boldsymbol{x}_u) \cdot \nabla_{\boldsymbol{w}}\boldsymbol{z}_y(\boldsymbol{x}_o)}{|\nabla_{\boldsymbol{w}}\boldsymbol{z}_y(\boldsymbol{x}_u)|^2} \leq t \wedge \frac{\nabla_{\boldsymbol{w}}\boldsymbol{z}_y(\boldsymbol{x}_u) \cdot \nabla_{\boldsymbol{w}}\boldsymbol{z}_y(\boldsymbol{x}_o)}{|\nabla_{\boldsymbol{w}}\boldsymbol{z}_y(\boldsymbol{x}_o)|^2} \leq t$$

where $t \in [-1, -\frac{1}{2})$. This situation is visualised in Figure 3c.

However, the possible causes of this case is more complicated than the above two.

- $y_u = y_o$, i.e. the two samples are from the same class. Suppose $\boldsymbol{x}_u$ is interchangeable with many other samples in the same class and $\boldsymbol{x}_o$ is not, this may indicate that $\boldsymbol{x}_o$ is an outlier of the class.

- $y_u \neq y_o$, i.e. the samples are from different classes. Since the annotated data in these two classes can not be learnt by the model at the same time, our guess is that the data is not completely compatible with the learning model, or the learning model may have a wrong bias for the task.

C.4 CONTRADICTORY SAMPLES IN PRACTICE

In the experiments showed in Section 3.2, we found the contradictory samples are relatively rare in MNIST and CIFAR. This might due to that the labels in benchmarks like MNIST and CIFAR are relatively clean. We believe that if the input signal is sampled uniformly, then the more complex the input samples are (e.g., colour images are more complex than black-and-write ones, high-resolution images are more complex than low-resolution ones) the less likely that the contradictory samples occur. This is due to that larger feature space as well as parameter space makes it less likely for two samples with different labels to have identical features or representations. As long as these two samples have different values on just one dimension, we cannot say they are strictly contradictory.

But, in practice, we believe that there are indeed specific cases that can introduce many contradictory signals. For example, we can do so by manually flipping the label, or systematically changing the data collector. Moreover, contradiction can also happen when models fit a good high-level representations for the samples. For example, two samples both contain a cow in the foreground and a grass land as the background, and one is labelled as "cow" while the other is labelled as "grass". In this case, the high-level feature combination is identical, i.e (cow, grass), but the mappings might cancel each other during the learning of the model.

# D FURTHER DETAILS ABOUT EXPERIMENTS IN SECTION 3

## D.1 AN EXTREME EXAMPLE OF EASY/HARD SAMPLES DUE TO INTERACTIONS

In this section, we illustrate an extreme case of interchangeable sample and another extreme case of contradictory samples. To simplify the task, we assume $K = 2$, i.e. a binary classification problem, and there are only two annotated samples in the training set, $(\boldsymbol{x}_u, y_u)$ and $(\boldsymbol{x}_u, y_o)$. Note that the inputs are identical, and we discuss $y_o = y_u$ and $y_o \neq y_u$ separately in the following.

**1.** $y_o = y_u$ In this case, the two annotated sample are actually identical, thus learning any one of them is equivalent to learning both. It is then straightforward to see that these two (identical) samples are interchangeable. Though they are identical, the one that appears later would be counted as an *easy* sample, since the updates on the other one automatically lead to good predictions for both.

**2.** $y_o \neq y_u$ In this case, since there are only two categories, learning one of them would cancel the update due to the other one. Suppose the model is being trained in an online learning fashion, and the two samples appear in turn. Then, learning $(\boldsymbol{x}_u, y_u)$ drags the model's prediction $\boldsymbol{q}(\boldsymbol{x}_u)$ towards $y_u$. On the other hand, learning $(\boldsymbol{x}_u, y_o)$ drags the model's prediction $\boldsymbol{q}(\boldsymbol{x}_u)$ towards the opposite direction $y_o$. In this way, the two samples are completely contradictory to each other, and they both are *difficult* samples, since the learning is unlikely to converge.

## D.2 FURTHER DETAILS ABOUT THE LEARNING DIFFICULTY CORRELATION EXPERIMENT

In this section, we illustrate more details about how we measure the learning difficulty of samples in datasets with varying sizes, as a supplement to Section 3.2. Let's suppose that the original training set $\mathbb{T}$ contains $N$ samples of $K$ classes. For a given subset size $X$, we construct $\frac{N}{X}$ proper subsets of $\mathbb{T}$, $\mathcal{T} \triangleq \{\mathbb{T}_1, \ldots, \mathbb{T}_{\frac{N}{X}}\}$, of which each contains $X$ samples drawn uniformly at random from all classes without replacement. To get the Pearson correlation coefficient $\rho$ between learning difficulty of samples on $\mathbb{T}$ and $\mathcal{T}$, we run the following Algorithm 1.

In Section 3.2, we set $N = 2000$ on MNIST. To further eliminate the effects from hyperparameters, we run Algorithm 1 on both MNIST and CIFAR-10 with varying learning rates as well as batch size. In these experiments, we train LeNet-5 on MNIST and ResNet-18 on CIFAR-10. The results and the corresponding hyperparameters are listed below:

1. this setting runs on MNIST with $N = 4096, X \in \{1, 4, 16, 64, 256, 1024\}$, learning rate is set to $0.1$, and batch size is $128$, the correlation results are given in Table 4 and are visualised in Figure 4;

**Algorithm 1:** Correlation between Learning Difficulty on Size $N$ and Size $X$

---

**Input:**    $\mathbb{T} = \{(\boldsymbol{x}_n, y_n)\}_{n=1}^N$    the training set
              $\boldsymbol{w}, f$             initial parameters, and the ANN to train
              $E$                 number of training epochs
              $\eta$                 learning rate

1   $\mathbb{L} \leftarrow \underbrace{\{0, \ldots, 0\}}_{N \text{ in total}}$ ;             `// List of sample learning difficulty on` $\mathbb{T}$

2   $\tilde{\mathbb{L}} \leftarrow \underbrace{\{0, \ldots, 0\}}_{N \text{ in total}}$ ;          `// Lists of sample learning difficulty on subsets`

3   $\boldsymbol{w}_0 \leftarrow \boldsymbol{w}$ ;                 `// Initialise the parameters to learn`

4   **for** $e \leftarrow 1$ **to** $E$ **do**

5      **for** $n \leftarrow 1$ **to** $N$ **do**

6          $\boldsymbol{q} \leftarrow f(\boldsymbol{x}_n; \boldsymbol{w}_0)$;

7          $l \leftarrow -\sum_{k=1}^K \mathbf{1}(k = y_n) \log(\boldsymbol{q}_k)$ ;          `// Cross-entropy loss`

8          $\mathbb{L}_n \leftarrow \mathbb{L}_n + l$;

9          $\boldsymbol{w}_0 \leftarrow \boldsymbol{w}_0 + \eta \times \nabla_{\boldsymbol{w}_0} l$;

10   **for** $j \leftarrow 1$ **to** $\frac{N}{X}$ **do**

11      $\boldsymbol{w}_j \leftarrow \boldsymbol{w}$ ;             `// Initialise the parameters for subset` $j$

12      **for** $e \leftarrow 1$ **to** $E$ **do**

13          **for** $n \leftarrow (j-1) \cdot X - 1$ **to** $j \cdot X$ **do**      `// enumerate over a subset of` $\mathbb{T}$

14             $\boldsymbol{q} \leftarrow f(\boldsymbol{x}_n; \boldsymbol{w}_j)$;

15             $l \leftarrow -\sum_{k=1}^K \mathbf{1}(k = y_n) \log(\boldsymbol{q}_k))$;

16             $\tilde{\mathbb{L}}_n \leftarrow \tilde{\mathbb{L}}_n + l$;

17             $\boldsymbol{w}_j \leftarrow \boldsymbol{w}_j + \eta \times \nabla_{\boldsymbol{w}_j} l$;

**Output:** The Pearson correlation coefficient $\rho$ between $\mathbb{L}$ and $\tilde{\mathbb{L}}$

---

2. this setting runs on MNIST with $N = 4096, X \in \{1, 4, 16, 64, 256, 1024\}$, learning rate is set to $0.001$, and batch size is 256, the correlation results are given in Table 5 and are visualised in Figure 5;

3. this setting runs on CIFAR10 with $N = 4096, X \in \{1, 4, 16, 64, 256, 1024\}$, learning rate is set to $0.1$, and batch size is 128, the correlation results are given in Table 6 and are visualised in Figure 6;

Note that the hyperparameter combinations in all the above settings are different from the experiment we ran for Figure 1. Therefore, we argue that the conclusion on learning difficulty correlations experiments consistently holds across different combinations of hyperparameters, $N$ and $X$.

| Size | 4096 | 1024 | 256 | 64 | 16 | 4 | 1 |
|------|------|------|------|------|------|------|------|
| 4096 | 1.0 | 0.8512 | 0.7282 | 0.5753 | 0.3865 | 0.2330 | 0.062 |
| 1024 | | 1.0 | 0.8206 | 0.6775 | 0.4731 | 0.2888 | 0.0658 |
| 256 | | | 1.0 | 0.7856 | 0.5853 | 0.3687 | 0.0858 |
| 64 | | | | 1.0 | 0.7169 | 0.4871 | 0.1566 |
| 16 | | | | | 1.0 | 0.6379 | 0.2303 |
| 4 | | | | | | 1.0 | 0.3720 |
| 1 | | | | | | | 1.0 |

Table 4: The Pearson correlation coefficients $\rho$ between learning difficulty on MNIST where $N = 4096, X \in \{1, 4, 16, 64, 256, 1024\}$, learning rate is set to $0.1$, and batch size is 128, i.e. setting 1. As shown in the table, larger gap between the sizes always lead to smaller $\rho$, which matches with our Prediction Prediction 1 illustrated in Section 3.2.

| Size | 4096 | 1024 | 256 | 64 | 16 | 4 | 1 |
|------|------|------|------|------|------|------|------|
| 4096 | 1.0 | 0.9096 | 0.7376 | 0.4065 | 0.2141 | 0.0695 | 0.0307 |
| 1024 | | 1.0 | 0.8830 | 0.5128 | 0.2707 | 0.0858 | 0.0498 |
| 256 | | | 1.0 | 0.6782 | 0.3486 | 0.0971 | 0.0605 |
| 64 | | | | 1.0 | 0.5898 | 0.1774 | 0.1598 |
| 16 | | | | | 1.0 | 0.6601 | 0.6023 |
| 4 | | | | | | 1.0 | 0.8414 |
| 1 | | | | | | | 1.0 |

Table 5: The Pearson correlation coefficients $\rho$ between learning difficulty on MNIST where $N = 4096, X \in \{1, 4, 16, 64, 256, 1024\}$, learning rate is set to $0.001$, and batch size is 256, i.e. setting 2. As shown in the table, larger gap between the sizes always lead to smaller $\rho$, which matches with our Prediction Prediction 1 illustrated in Section 3.2.

| Size | 4096 | 1024 | 256 | 64 | 16 | 4 | 1 |
|------|------|------|------|------|------|------|------|
| 4096 | 1.0 | 0.8433 | 0.5772 | 0.2805 | 0.1840 | 0.0628 | 0.0131 |
| 1024 | | 1.0 | 0.7377 | 0.4142 | 0.2595 | 0.0950 | 0.0254 |
| 256 | | | 1.0 | 0.5409 | 0.3979 | 0.1352 | 0.0345 |
| 64 | | | | 1.0 | 0.4082 | 0.1493 | 0.0427 |
| 16 | | | | | 1.0 | 0.1844 | 0.0673 |
| 4 | | | | | | 1.0 | 0.0912 |
| 1 | | | | | | | 1.0 |

Table 6: The Pearson correlation coefficients $\rho$ between learning difficulty on CIFAR10 where $N = 4096, X \in \{1, 4, 16, 64, 256, 1024\}$, learning rate is set to $0.1$, and batch size is 128, i.e. setting 3. As shown in the table, larger gap between the sizes always lead to smaller $\rho$, which matches with our Prediction Prediction 1 illustrated in Section 3.2.

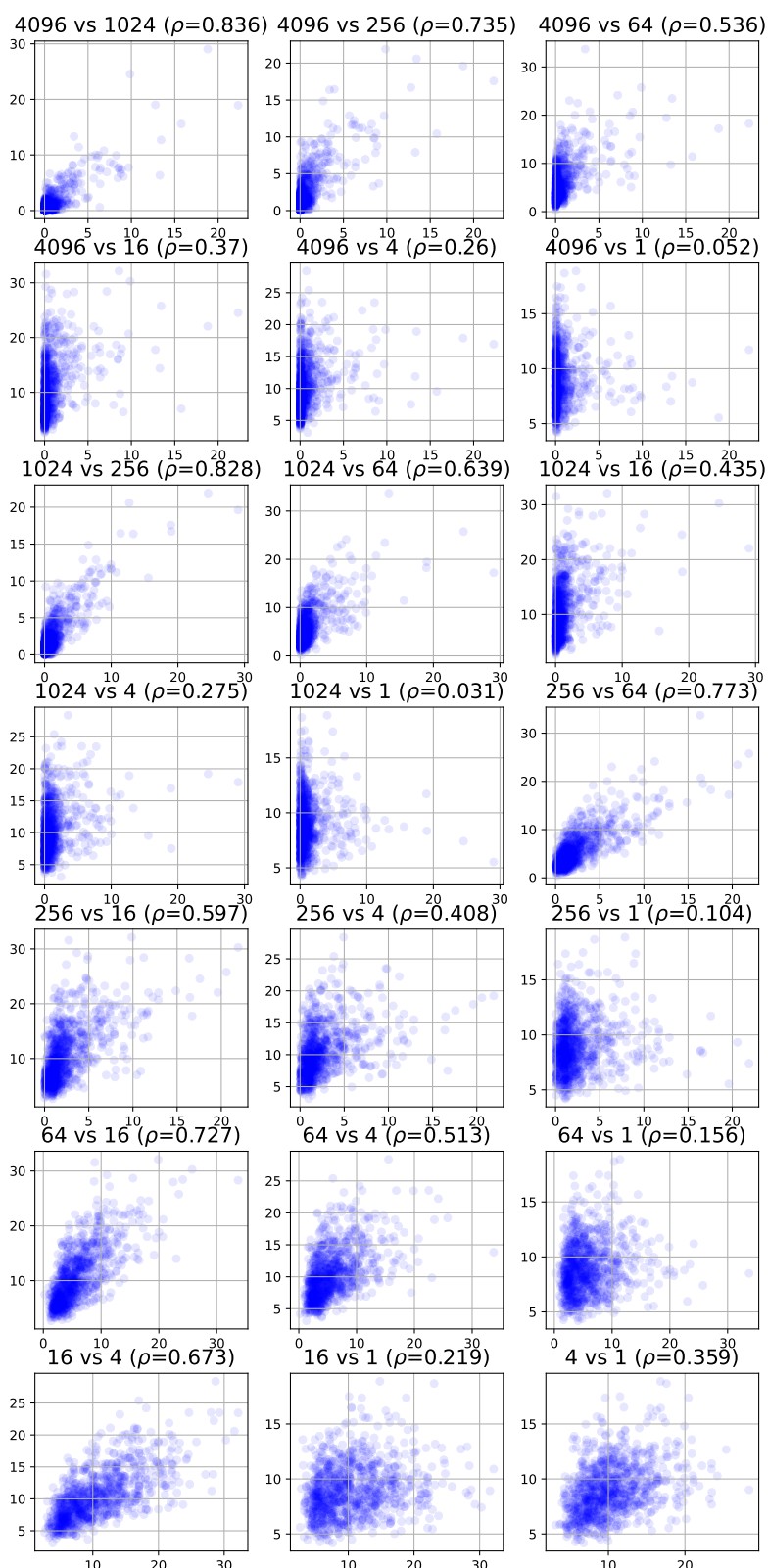

Figure 4: Visualisation of the correlation between learning difficulty on MNIST where $N = 4096$, $X \in \{1, 4, 16, 64, 256, 1024\}$, learning rate is set to $0.1$, and batch size is $128$, i.e. setting 1. Note that we plot only the correlation between $1000$ samples in order to keep the plots readable.

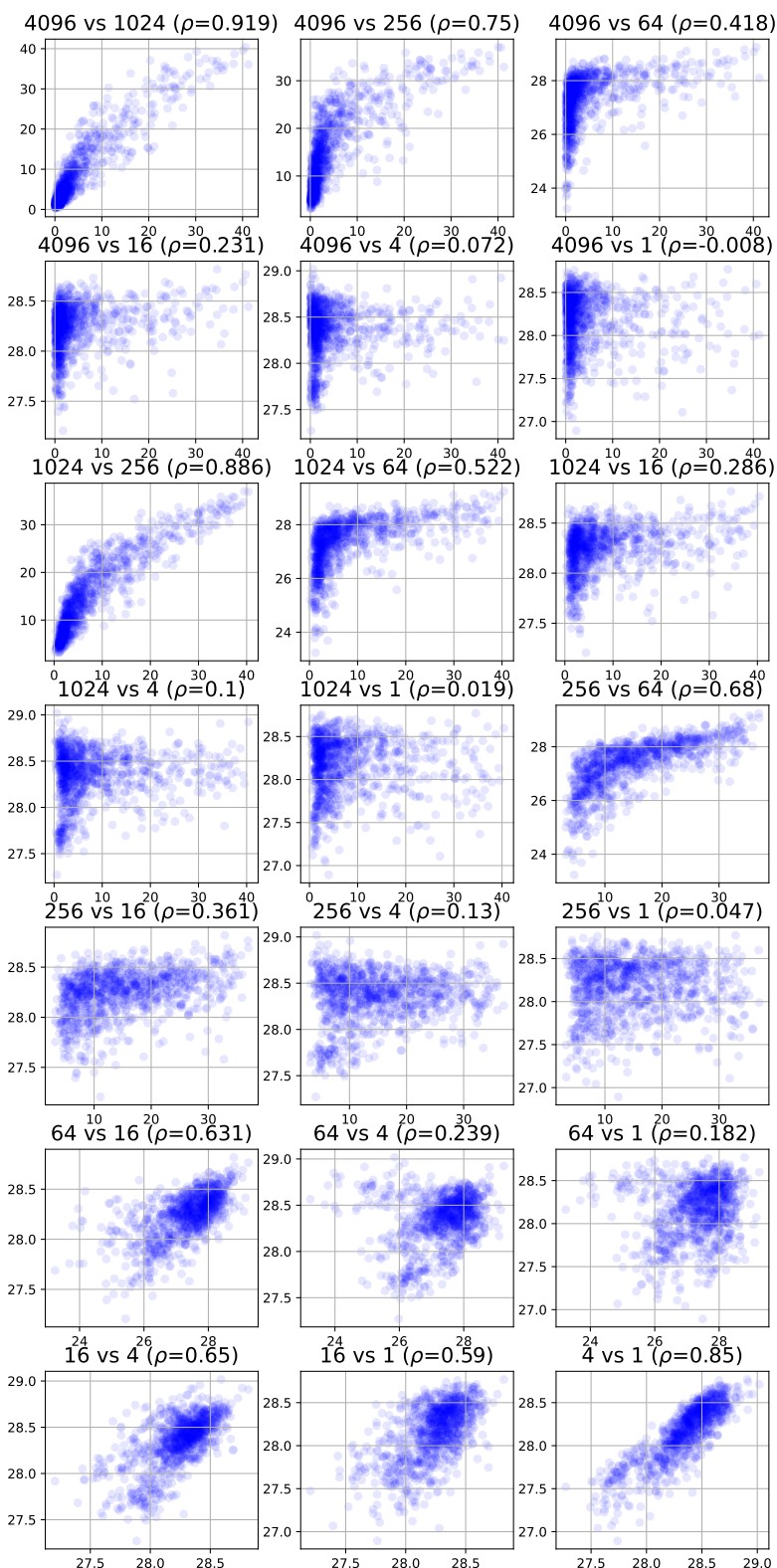

Figure 5: Visualisation of the correlation between learning difficulty on MNIST where $N = 4096$, $X \in \{1, 4, 16, 64, 256, 1024\}$, learning rate is set to $0.001$, and batch size is $256$, i.e. setting 2. Note that we plot only the correlation between $1000$ samples in order to keep the plots readable.

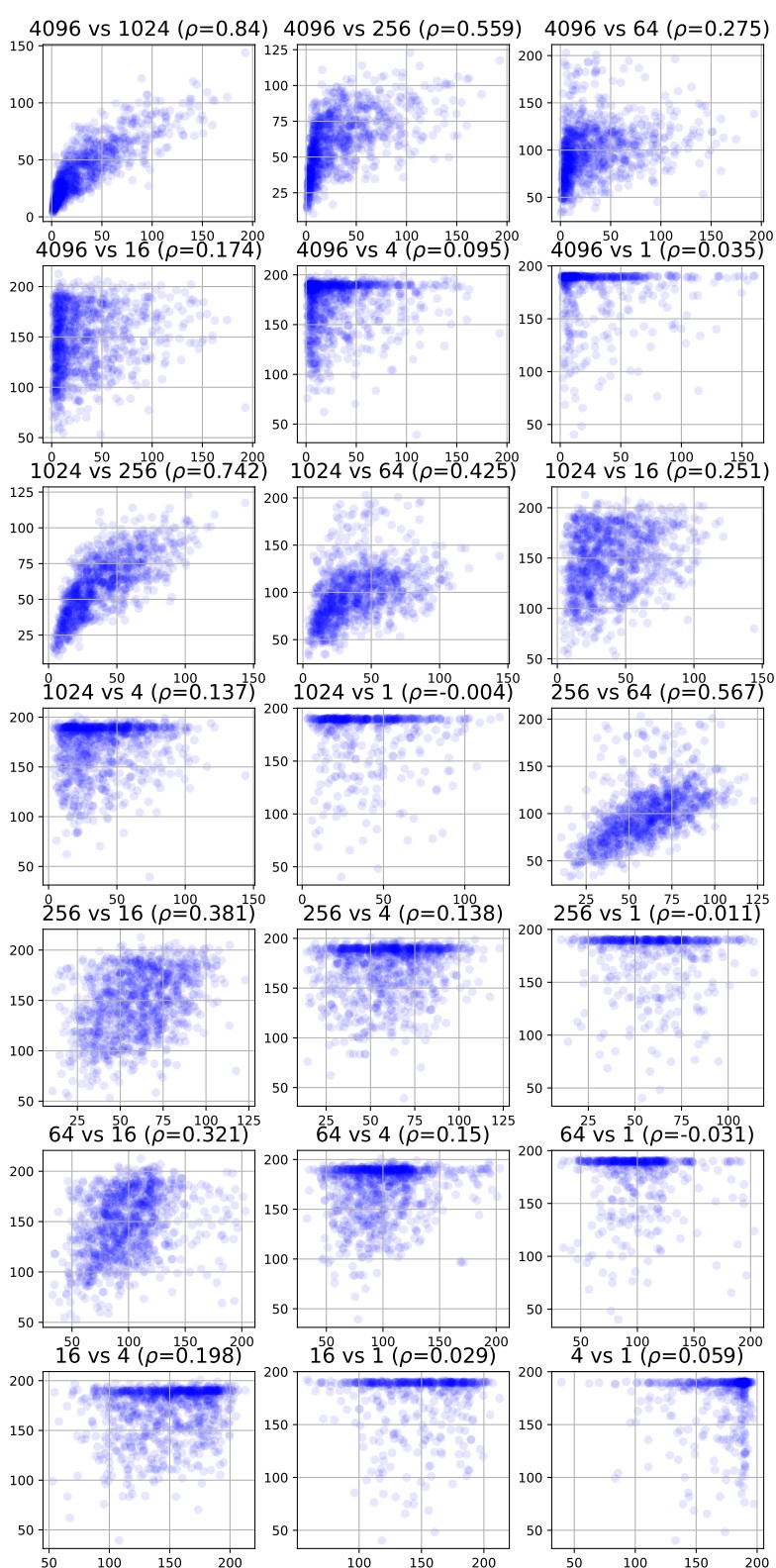

Figure 6: Visualisation of the correlation between learning difficulty on CIFAR10 where $N = 4096, X \in \{1, 4, 16, 64, 256, 1024\}$, learning rate is set to $0.1$, and batch size is $128$, i.e. setting 3. Note that we plot only the correlation between 1000 samples in order to keep the plots readable.

### D.3 FURTHER DETAILS ABOUT THE LEARNING DIFFICULTY CONTROL EXPERIMENT

In this section, we provide more details about the learning difficulty control experiment illustrated in Section 3.2. We first show the centroids of largest cluster obtained through farthest point clustering, of which more details can be found in Appendix D.5, of each class in MNIST and CIFAR10 in Figure 7 and Figure 8 respectively.

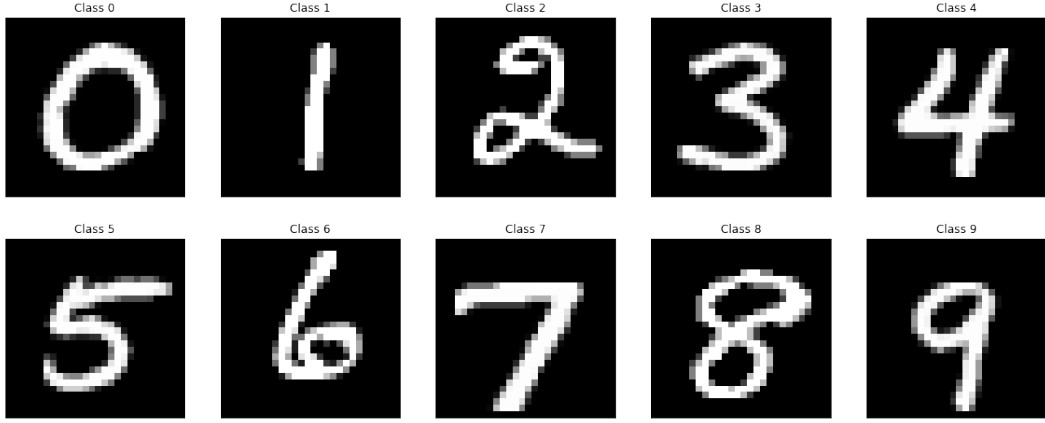

Figure 7: Target samples of each class in MNIST found by FPC with lpNTK as the similarity metric.

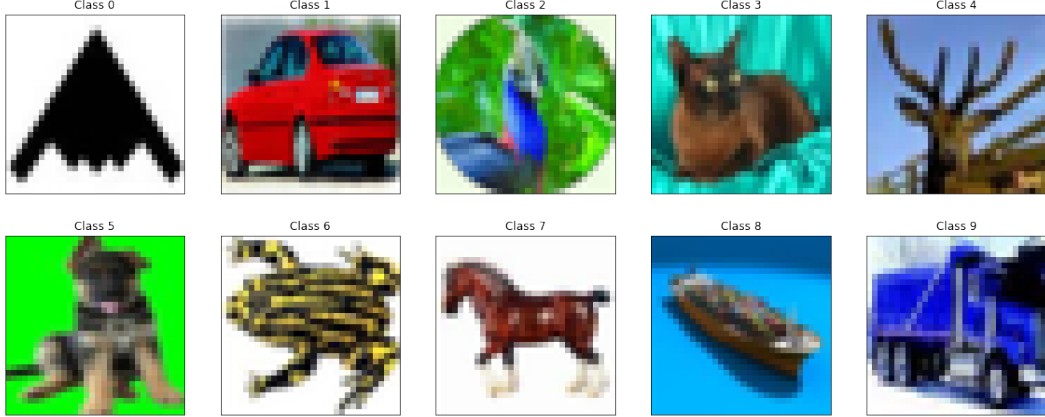

Figure 8: Target samples of each class in CIFAR10 found by FPC with lpNTK as the similarity metric.

Then, we show the samples that are the most interchangeable to the targets, thus make the targets easier to learn, on MNIST and CIFAR10 in Figure 9 and Figure 10 respectively. It is worth noting that the most interchangeable samples to the targets found by lpNTK are not necessarily the most similar samples to the targets in the input space. Take the "car" class in CIFAR10 (the second row of Figure 10) for example, the target is a photo of a **red** car from the **left back** view, and the third most interchangeable sample to it is a photo of a **yellow** from the **front right** view. At the same time, the fourth and fifth most interchangeable samples are both photos of **red** cars, thus they have shorter distance to the targets in the input space due to the large red regions in them. However, under the lpNTK measure, the fourth and fifth red cars are less interchangeable/similar to the target red car than the third yellow car. This example shows that our lpNTK is significantly different from the similarity measures in the input space.

We also show the samples that are medium interchangeable to the targets, thus make the targets medium easy to learn, on MNIST and CIFAR10 in Figure 11 and Figure 12 respectively.

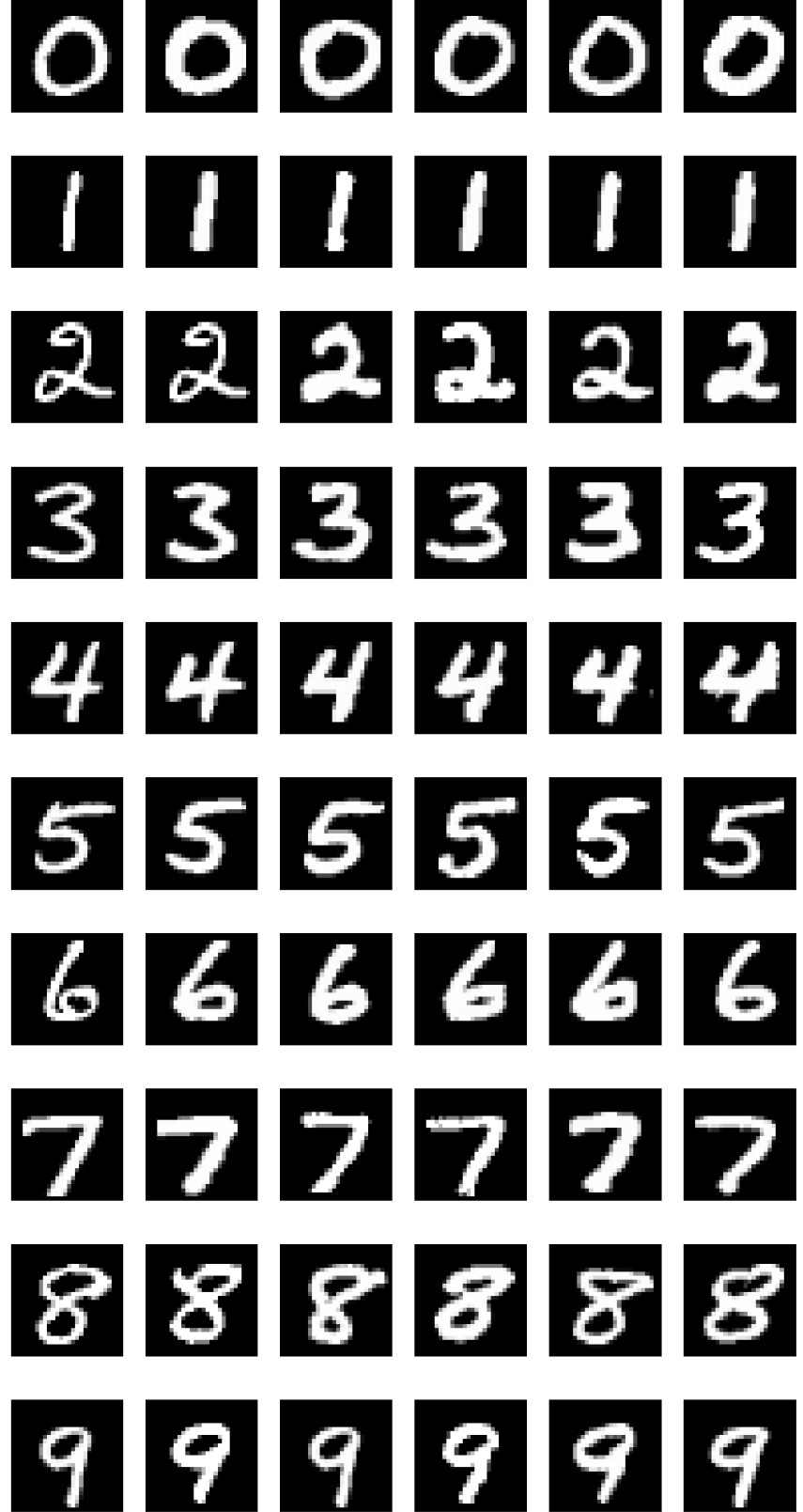

Figure 9: Samples that are the most interchangeable to the targets in MNIST. The leftmost column is the target samples that are also shown in Figure 7. The remaining 5 columns are the most interchangeable samples to the targets in each class.

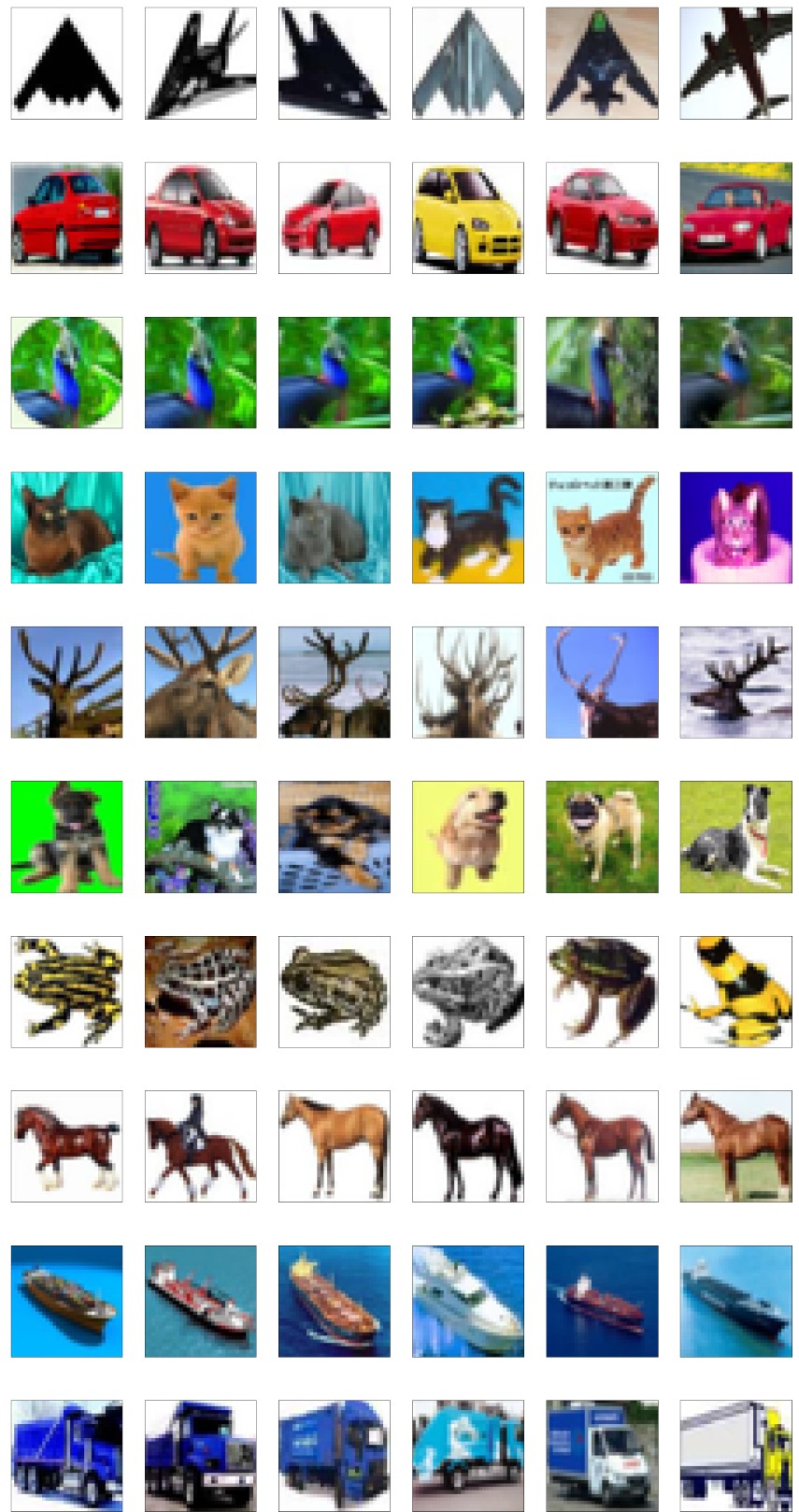

Figure 10: Sample that are the most interchangeable to the targets in CIFAR10. The leftmost column is the target samples that are also shown in Figure 8. The remaining 5 columns are the most interchangeable samples to the targets in each class.

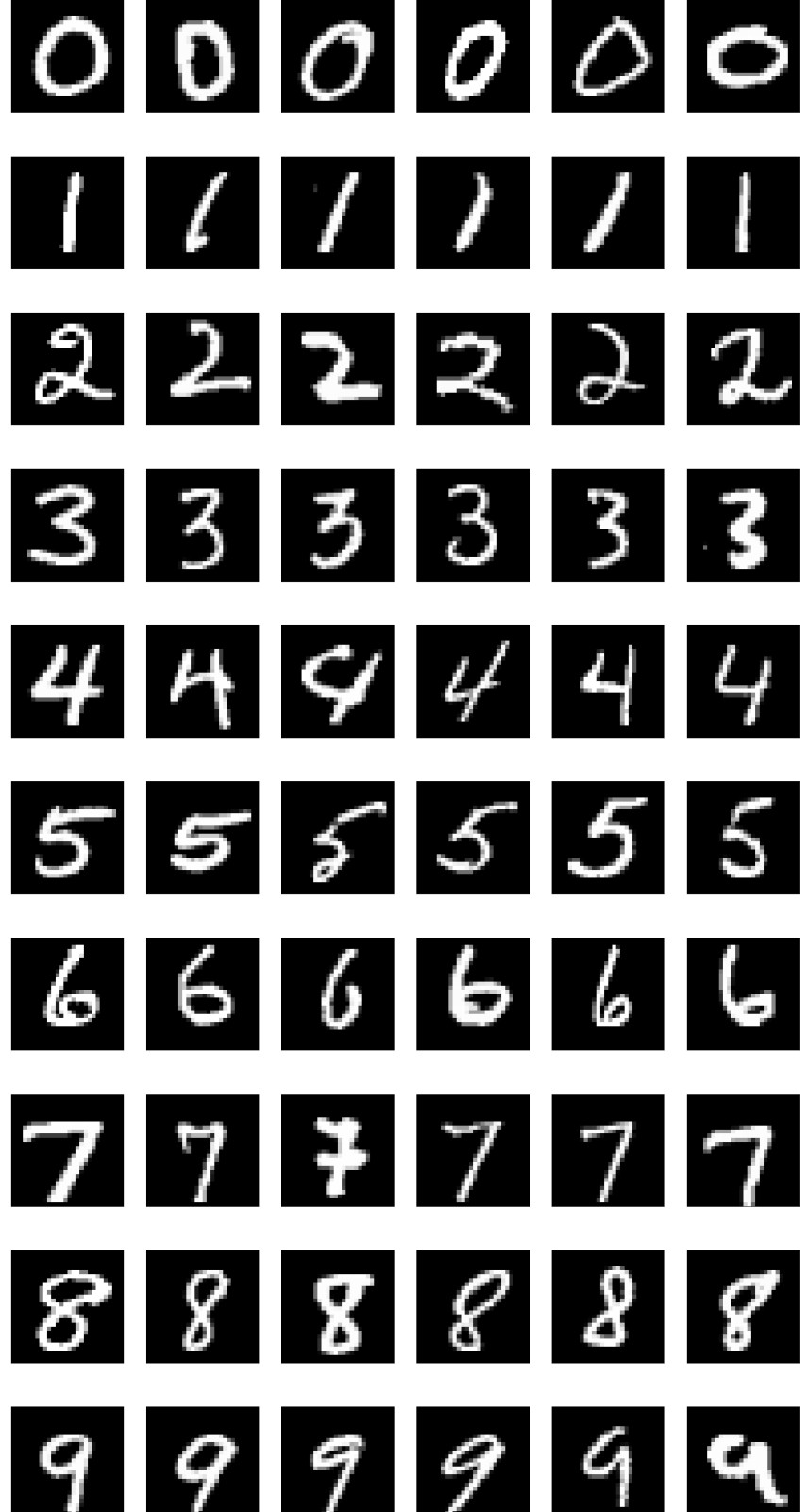

Figure 11: Samples that are the medium interchangeable to the targets in MNIST. The leftmost column is the target samples that are also shown in Figure 7. The remaining 5 columns are the medium interchangeable samples to the targets in each class.

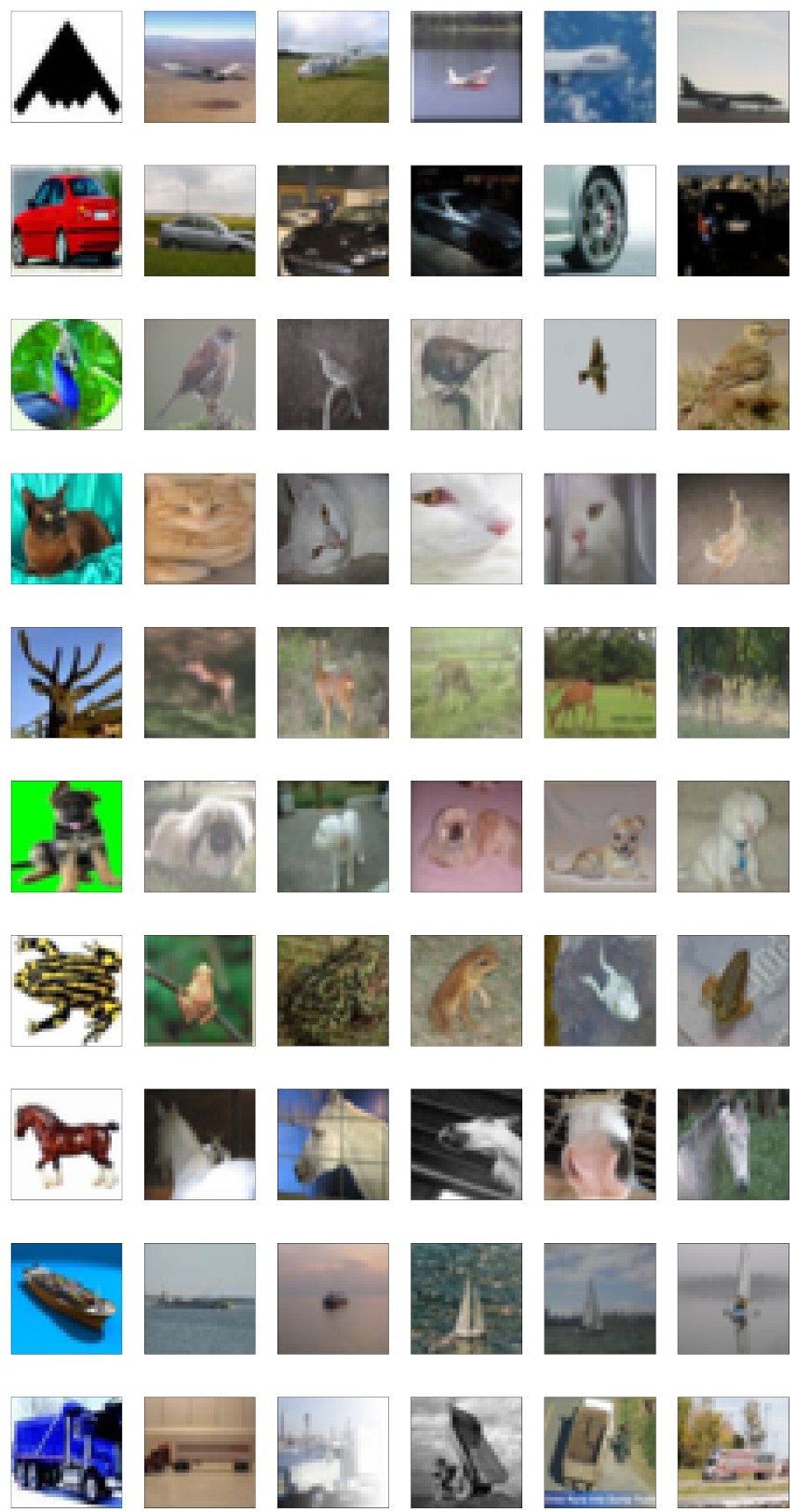

Figure 12: Sample that are the medium interchangeable to the targets in CIFAR10. The leftmost column is the target samples that are also shown in Figure 8. The remaining 5 columns are the medium interchangeable samples to the targets in each class.

Finally, we show the samples that are the most non-interchangeable to the targets, thus make the targets hard to learn, on MNIST and CIFAR10 in Figure 13 and Figure 14 respectively. We observed that these samples from the tail clusters are more likely to share features appearing in other classes. An example is the class "6" shown in the seventh row of Figure 13. As can be seen there, the second to fourth most non-interchangeable to the target look more like "4" than the others. Inspired by this observation, we argue that samples in the tail classes should be annotated with multiple labels rather than a single label, since they may contain similar "feature" from other classes. However, limited to the scope by this work, we leave this question to the future works.

### D.4 FURTHER DETAILS ABOUT THE FORGETTING EVENT PREDICTION EXPERIMENT

In this section, we illustrate more details about how we predict the forgetting events with our lpNTK, as a supplement to Section 3.3. Let's suppose at training iteration $t$, the parameter of our model $\boldsymbol{w}^t$ is updated with a batch of samples $\mathbb{B}^t = \{(\boldsymbol{x}_i^t, y_i^t)\}_{i=1}^B$ of size $B$, thus becomes $\boldsymbol{w}^{t+1}$. Then, at the next iteration $t + 1$, we update the model parameters $\boldsymbol{w}^{t+1}$ with another batch of samples $\mathbb{B}^{t+1} = \{(\boldsymbol{x}_i^{t+1}, y^{t+1})_i\}_{i=1}^B$ to $\boldsymbol{w}^{t+2}$. We found that the predictions on a large number of samples at the beginning of training are almost uniform distributions over classes, thus the prediction error term in Equation 2 is likely to be $[-\frac{1}{K}, \dots, 1 - \frac{1}{K}, \dots, -\frac{1}{K}]^\mathsf{T}$. To give a more accurate approximation to the change of predictions at the beginning of training, we therefore replace the $\boldsymbol{s}(y)$ in Equation 3 with $\tilde{\boldsymbol{s}}(y) = [-1, \dots, \underbrace{K-1}_{y-\text{th entry}}, \dots, -1]^\mathsf{T}$ and denote this variant of our lpNTK as $\tilde{\kappa}$. We then propose the following Algorithm 2 to get the confusion matrix of the performance of predicting forgetting events with $\tilde{\kappa}$.

---

**Algorithm 2:** Predict forgetting events with a variant of lpNTK $\tilde{\kappa}$

**Input:** $\mathbb{B}^t = \{(\boldsymbol{x}_i^t, y_i^t)\}_{i=1}^B$      the batch of samples at $t$
$\mathbb{B}^{t+1} = \{(\boldsymbol{x}_i^{t+1}, y^{t+1})_i\}_{i=1}^B$    the batch of samples at $t + 1$
$\boldsymbol{w}^t, \boldsymbol{w}^{t+1}, \boldsymbol{w}^{t+2}$      the saves parameters at time-step $t, t + 1$, and $t + 2$
$\eta$      learning rate

1   $\mathbb{F} \leftarrow \underbrace{\{0, \dots, 0\}}_{B \text{ in total}}$ ;      // List to record whether forgetting events happen

2   $\hat{\mathbb{F}} \leftarrow \underbrace{\{0, \dots, 0\}}_{B \text{ in total}}$ ;      // List to record the predicted forgetting events

3   **for** $i \leftarrow 1$ **to** $B$ **do**
4     **if** $\arg\max \boldsymbol{q}^{t+1}(\boldsymbol{x}_i^t; \boldsymbol{w}^{t+1}) = y_i^t \wedge \arg\max \boldsymbol{q}^{t+2}(\boldsymbol{x}_i^t; \boldsymbol{w}^{t+2}) \neq y_i^t$ **then**
5       $\lfloor \mathbb{F}_i \leftarrow 1$ ;      // Forgetting event happens
6     $\Delta q_i \leftarrow 0$;
7     **for** $j \leftarrow 1$ **to** $B$ **do**
8       $\lfloor \Delta q_i \leftarrow \Delta q_i + \tilde{\kappa}(\boldsymbol{x}_i^t, \boldsymbol{x}_j^{t+1}; \boldsymbol{w}^{t+1})$;

9     $\Delta \boldsymbol{q}_i \leftarrow \Delta \boldsymbol{q}_i \cdot \begin{bmatrix} -1 \\ \dots \\ K-1 \\ \dots \\ -1 \end{bmatrix}$ where $K - 1$ is the $y_i^t$-th element;

10    **if** $\arg\max \boldsymbol{q}^{t+1}(\boldsymbol{x}_i^t; \boldsymbol{w}^{t+1}) = y_i^t \wedge \arg\max [\boldsymbol{q}^{t+1}(\boldsymbol{x}_i^t; \boldsymbol{w}^{t+1}) + \Delta \boldsymbol{q}_i] \neq y_i^t$ **then**
11      $\lfloor \hat{\mathbb{F}}_i \leftarrow 1$ ;      // Predicted forgetting events

**Output:** The precision, recall, and F1-score of $\hat{\mathbb{F}}$ with $\mathbb{F}$ as ground truth

---

### D.5 FARTHEST POINT CLUSTERING WITH LPNTK

In this section, we illustrate more details about how we cluster the training samples based on farthest point clustering (FPC) and lpNTK. Suppose that there are $N$ samples in the training set,

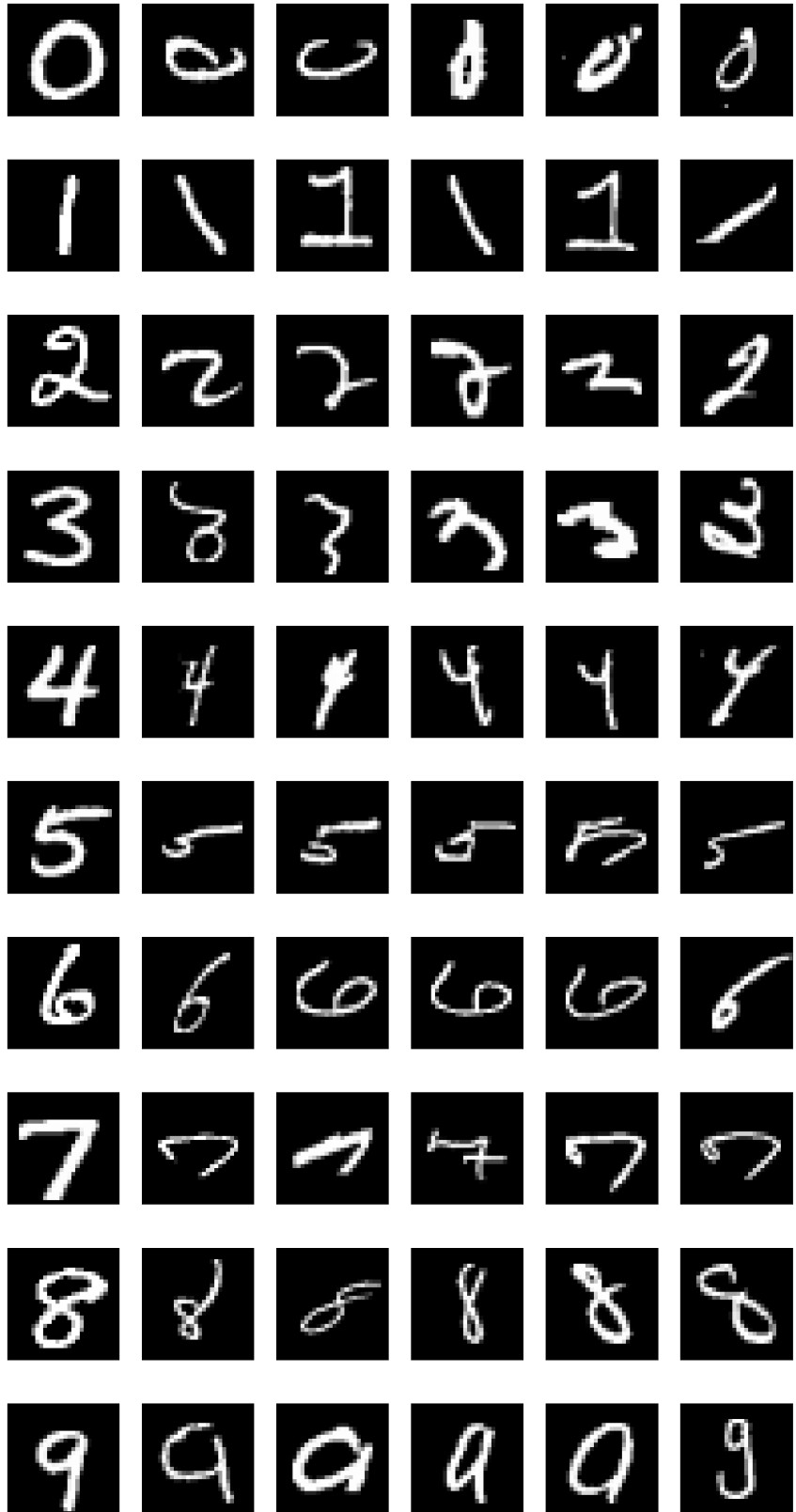

Figure 13: Samples that are the most non-interchangeable to the targets in MNIST. The leftmost column is the target samples that are also shown in Figure 7. The remaining 5 columns are the most non-interchangeable samples to the targets in each class.

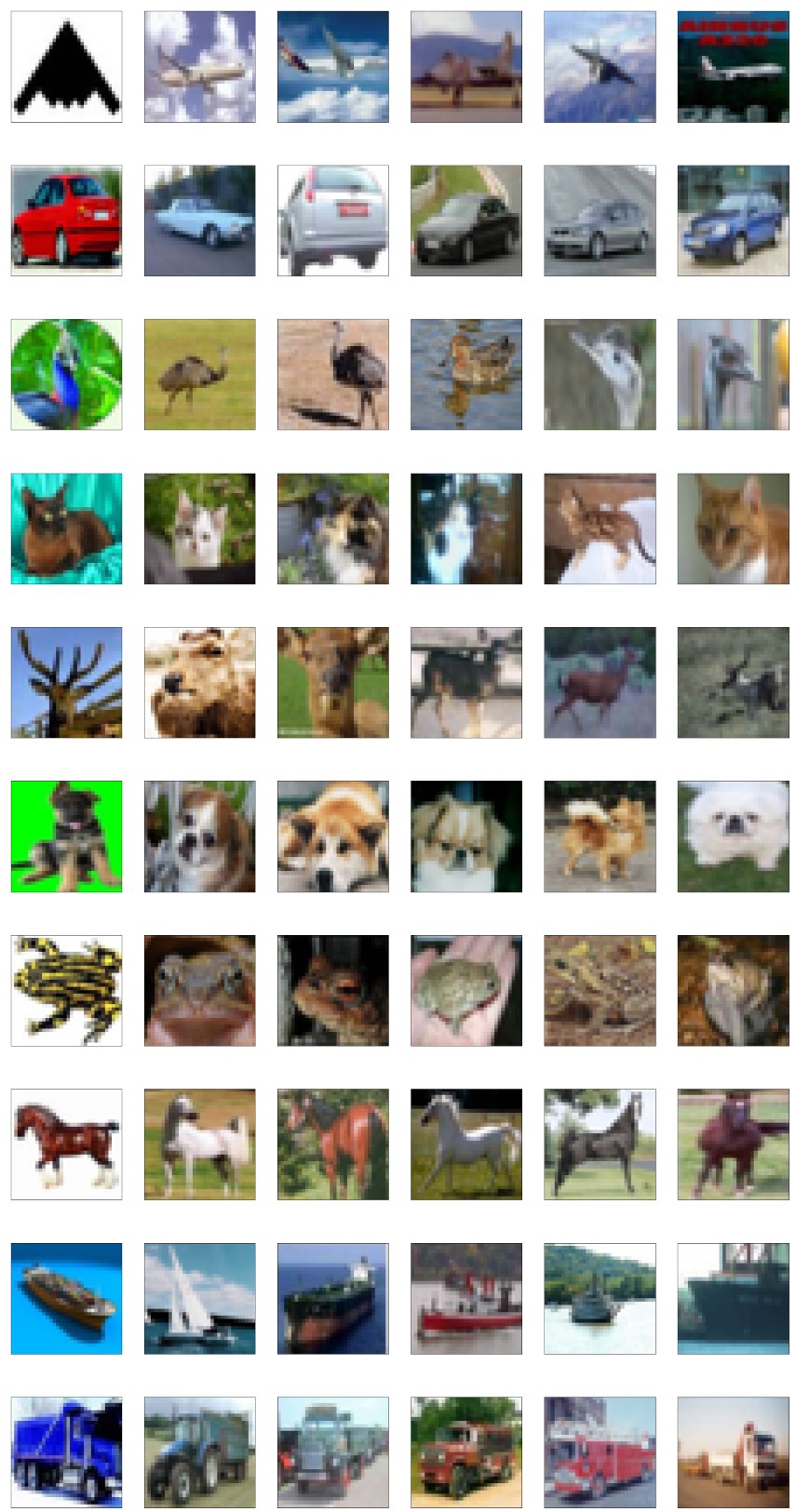

Figure 14: Sample that are the most non-interchangeable to the targets in CIFAR10. The leftmost column is the target samples that are also shown in Figure 8. The remaining 5 columns are the most non-interchangeable samples to the targets in each class.

$\{(\boldsymbol{x}_i, y_i)\}_{i=1}^N$, and the lpNTK between a pair of samples is denoted as $\kappa((\boldsymbol{x}, y), (\boldsymbol{x}', y'))$. The FPC algorithm proposed by Gonzalez (1985) running with our lpNTK is demonstrated in Algorithm 3.

---

**Algorithm 3:** Farthest Point Clustering with lpNTK

---

**Input:** Dataset $\{(\boldsymbol{x}_i, y_i)\}_{i=1}^N$, The number of centroids $M$
**Output:** Cluter centroids $\{c_1, \ldots, c_M\}$, LIst of samples indices in all clusters: $\{\mathbb{L}_1, \ldots, \mathbb{L}_M\}$

1  $N_c \leftarrow 1$ ;            // Number of established clusters
2  $c_1 \leftarrow \arg\max_i \kappa((\boldsymbol{x}_i, y_i), (\boldsymbol{x}_i, y_i))$ ;    // Centroid of the first cluster
3  $\mathbb{L}_1 \leftarrow \{i \neq c_1\}_{i=1}^N$ ;    // List of sample indices in the first clsuter
4  **while** $N_c < M$ **do**
5      $d \leftarrow \infty, c_{new} \leftarrow -1$ ;    // 1.  Find the controid for the new cluster
6      **for** $j \leftarrow 1$ **to** $N_c$ **do**
7          **if** $\min_{j' \in \mathbb{L}_j} \kappa((\boldsymbol{x}_{c_j}, y_{c_J}), (\boldsymbol{x}_{j'}, y_{j'})) < d$ **then**
8              $d \leftarrow \min_{j' \in \mathbb{L}_j} \kappa((\boldsymbol{x}_{c_j}, y_{c_J}), (\boldsymbol{x}_{j'}, y_{j'}))$;
9              $c_{new} \leftarrow \arg\min_{j' \in \mathbb{L}_j} \kappa((\boldsymbol{x}_{c_j}, y_{c_J}), (\boldsymbol{x}_{j'}, y_{j'}))$;
10     $N_c \leftarrow N_c + 1, c_{N_c} \leftarrow c_{new}, L_{N_c} \leftarrow \varnothing$ // 2.  Move samples that are closer to the new centroid to the new cluster
11     **for** $j \leftarrow 1$ **to** $N_c - 1$ **do**
12         **for** $i \in \mathbb{L}_j$ **do**
13             **if** $\kappa((\boldsymbol{x}_i, y_i), (\boldsymbol{x}_{c_j}, y_{c_j})) < \kappa((\boldsymbol{x}_i, y_i), (\boldsymbol{x}_{c_{N_c}}, y_{c_{N_c}}))$ **then**
14                 $\mathbb{L}_j \leftarrow \mathbb{L}_j \setminus \{i\}$;
15                 $\mathbb{L}_{N_c} \leftarrow \mathbb{L}_{N_c} \cup \{i\}$;

---

To show that the distribution of clusters sizes are heavily *long-tailed* on both MNIST and CIFAR-10, i.e. there are a few large clusters and many small clusters, we use $M = 10\% \times N$ here. We sort the clusters from FPC first by size, then show the sizes of the top-30 on MNIST and CIFAR10 in Figure 15. It is straightforward to see that most of the samples are actually in the head cluster, i.e. the largest cluster. There are a further $5 - 20$ clusters containing more than one sample, then the remaining clusters all contain just a single sample. This indicates that the cluster distribution is indeed long-tailed.

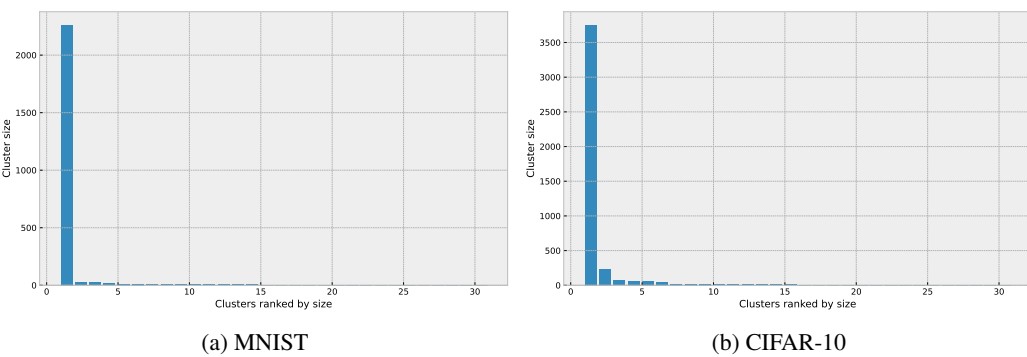

(a) MNIST                         (b) CIFAR-10

Figure 15: Number of samples in the top-30 clusters ranked by sizes on MNIST and CIFAR-10. The number of clusters equals to $10\%$ of the original training set's size, and both used categories indexed 0 as an example.

## E   REDUNDANT SAMPLES AND POISONING SAMPLES

### E.1   REDUNDANT SAMPLES

Suppose there are two samples $\boldsymbol{x}_1$ and $\boldsymbol{x}_2$ from the same class, and their corresponding gradient vectors in the lpNTK feature space are drawn as two solid arrows in Figure 16. The $\boldsymbol{x}_1$ can be

further decomposed into two vectors, $x_1'$ which has the same direction as $x_2$ and $x_1''$ whose direction is perpendicular to $x_2$, and both are drawn as dashed arrows in Figure 16. Since $x_1'$ is larger than $x_2$, this means that a component of $x_1$ is larger than $x_2$ on the direction of $x_2$. Further, this shows that learning $x_1$ is equivalent to learning $x_2$ (times some constant) and a hypothetical samples that is orthogonal to $x_2$. In this case, the sample $x_2$ is said to be **redundant**.

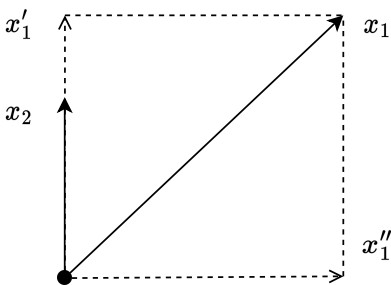

Figure 16: A hypothetical redundant sample ($x_2$) example.

### E.2  POISONING SAMPLES

Data poisoning refers to a class of training-time adversarial attacks Wang et al. (2022): the adversary is given the ability to insert and remove a bounded number of training samples to manipulate the behaviour (e.g. predictions for some target samples) of models trained using the resulted, poisoned data. Therefore, we use the name of "poisoning samples" here rather than "outliers" or "anomalies" [5] to emphasise that those samples lead to worse generalisation.

> More formally, for a given set of samples $\mathbb{T}$, if the averaged test accuracy of the models trained on $\mathbb{X} \subset \mathbb{T}$ is statistically significantly greater than the models trained on $\mathbb{T}$, the samples in $\mathbb{T} \setminus \mathbb{X}$ are defined as poisoning samples.

To better illustrate our findings about poisoning samples, we first illustrate a hypothetical FPC results, as shown in Figure 17. There are 5 resulted clusters, and the number of samples in each cluster is $7, 2, 1, 1, 1$. Back to the three types of relationships defined in Section 3, samples in the same cluster (e.g. the blue dots) would be more likely to be interchangeable with each other, and samples from different clusters (e.g. the yellow, green, and red dots) would be more likely to be unrelated or contradictory.

Although those samples in the tail did not seem to contribute much to the major group, the interesting thing is that they are **not always** poisoning. That is, the poisoning data are inconsistent over the varying sizes of the pruned dataset.

Let's denote the original training set as $\mathbb{T}$, and the pruned dataset as $\mathbb{X}$. When a large fraction of data can be kept in the pruned dataset, e.g. $|\mathbb{X}|/|\mathbb{T}| > 85\%$, removing some samples that are more interchangeable with the others, or equivalently from the head cluster, would benefit the test accuracy, as demonstrated in Section 4.2. That is, in such case, the poisoning samples are the samples in the largest cluster, e.g. the blue dots in Figure 17.

However, where only a small fraction of data can be kept, e.g. $5\%$ or $1\%$, we find that the poisoning samples are the samples from the tail clusters, e.g. the yellow/green/red dot in Figure 17. This finding matches with the conclusions from Sorscher et al. (2022). To verify it, we first randomly sample $5\%$ from the MNIST training set, and put them all together in a training set $\mathbb{T}_1$. In the meantime, we sample the same amount of data only from the *largest* cluster obtained on MNIST, and put them all together in another training set $\mathbb{T}_2$. We then compare the generalisation performance of LeNet models trained with $\mathbb{T}_1$ and $\mathbb{T}_2$, and we try two fractions, $1\%$ and $5\%$. The results are shown in the following Figure 18.

---

[5]Many attempts have been made to define outliers or anomaly in statistics and computer science, but there is no common consensus of a formal and general definition.

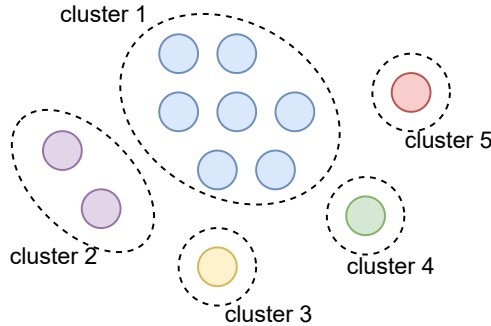

Figure 17: An example of farthest point clustering result.

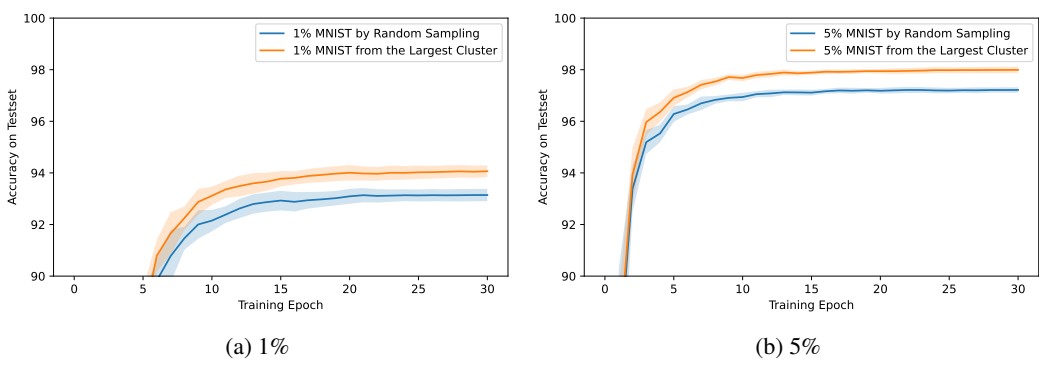

(a) 1%          (b) 5%

Figure 18: Test accuracy over training epochs of models trained with $X\%$ data sampled with different strategies. The lines are averaged across 10 different runs, and the shadow areas show the corresponding standard deviation. The plots show that sampling from the largest FPC cluster can lead to higher generalisation performance when only a small fraction of data can be kept in the pruned training sets.

As shown in Figure 18, in both $1\%$ and $5\%$ cases, keeping only the samples from the largest cluster obtained via FPC with lpNTK lead to higher test accuracy than sampling from the whole MNIST training set. This verifies our argument that the most interchangeable samples are more important for generalisation when only a few training samples can be used for learning.

Lastly, we want to give a language learning example to further illustrate the above argument. Let's consider the plural nouns. We just need to add -s to the end to make regular nouns plural, and we have a few other rules for the regular nouns ending with e.g. -s, -sh, or -ch. Whereas, there is no specific rule for irregular nouns like child or mouse. So, the irregular nouns are more likely to be considered as samples in the tail clusters, as they are more dissimilar to each other. For an English learning beginner, the most important samples might be the regular nouns, as they can cover most cases of changing nouns to plurals. However, as one continues to learn, it becomes necessary to remember those irregular patterns in order to advance further. Therefore, suppose the irregular patterns are defined as in the tail clusters, they may not always be poisoning for generalisation performance.

