# OpenReview forum: "lpNTK: Better Generalisation with Less Data via Sample Interaction During Learning"
_ICLR.cc/2024/Conference — ICLR 2024 poster_

### Official Review · Reviewer_9biR · 2023-11-01

**Soundness:** 3 good
**Presentation:** 3 good
**Contribution:** 3 good
**Rating:** 8
**Confidence:** 3

**Summary:**

This paper aims to study the interaction between samples in supervised learning from a learning dynamics viewpoint. To that end, the authors propose a the labeled pseudo Neural Tangent Kernel (lpNTK), a new adaptation of the NTK which explicitly incorporates label information into the kernel. First, lpNTK is shown to asymptotically converge to the empirical Neural Tangent Kernel (eNTK). Then, the authors demonstrate how lpNTK helps to understand phenomena such as identifying and interpreting easy/hard examples and forgetting events. Finally, the paper shows a case study in which lpNTK is used to improve the generalization performance of neural nets in image classification via pruning and de-biasing.

**Strengths:**

- The paper's main conceptual contribution of incorporating label information into the NTK is novel and interesting!
- The connection to sample difficulty and forgetting events is an interesting use-case for the lpNTK and allows us to connect sample difficulty with training dynamics.
- I found it pretty interesting that lpNTK allows for pruning and simultaneous utility improvements. This could open up another connection to selective prediction (learning a high-confidence subset of the distribution).
- Related work appears sufficiently discussed.

**Weaknesses:**

- Should Eq (1) contain the sign operator wrapped around the last factor? In Section 2.2, it is stated that the paper does not use the whole error term but only the sign as shown in Eq (2). I am also not sure I was able to follow the explanation as to why the magnitudes should not matter. Could the magnitude be useful for ranking?
- Although Figure 2 and the displayed evolution of distinct FPC clusters is insightful, I was wondering whether additional visualizations of the lpNTK would have been possible?
- The results in Table 1 showing the forgetting events don't seem particularly impressive (especially on MNIST). Am I missing something here?
- It would have been great if the authors had provided a visualization of some of the pruned samples. Are there any patterns present in pruned examples in particular?
- It is evident that the authors tried really hard to fit the paper into the page limit. There are formatting artifacts in terms of very little vspace throughout the paper, especially towards the end of the paper (pages 7-9). I would strongly encourage the authors to reduce dangling words instead of resorting to these very evident formatting tricks.

**Questions:**

- Have the authors considered drawing connections between interchangeable and contradictory samples and forging / machine unlearning [1], i.e. by training on a specific point we could plausibly say that we have also optimized for another datapoint?

Other questions embedded in Weaknesses above.

I am willing to increase my score as part of the discussion phase if the authors can address my concerns.

References:

[1] Thudi, Anvith, et al. "On the necessity of auditable algorithmic definitions for machine unlearning." 31st USENIX Security Symposium (USENIX Security 22). 2022.

---

> ### Author Response · Authors · 2023-11-19
> **Reply to reviewer 9biR (1/2)**
>
> We thank the reviewer for identifying the novel and interesting topic of our work. We answer the reviewer’s concerns one-by-one below.
>
> 1. Why error magnitude is not so important
>
> > Should Eq (1) contain the sign operator wrapped around the last factor? In Section 2.2, it is stated that the paper does not use the whole error term but only the sign as shown in Eq (2). I am also not sure I was able to follow the explanation as to why the magnitudes should not matter. Could the magnitude be useful for ranking?
>
> We thank the reviewer for pointing out this. Actually, Eq (1) should not contain the sign operator, as it comes from the definition of gradient descent and Taylor expansion. For the magnitude part, it is true that this value is useful for ranking at the early stage of training, which is what most of the learning dynamic-based methods use, like forgetting score, C-score, etc. However, our method considers a different stage during training, i.e., a fully-trained model (but before overfitting). At this stage, the training accuracy on many samples is low, which means for most of the samples, the magnitudes of |p-q| are quite small. Thus, at this stage, the |p-q| are in the similar value for most of the samples. Hence at this stage, using the sign operator is a good approximation of the original p-q. In summary, we can consider Eq (1) in this way: $\Delta q = \eta\cdot A\cdot K\cdot sign(p-q)\cdot |p-q|$. We merge $K$ and $sign(p-q)$ to create an approximation.
>
> Moreover, an extreme example could be two identical samples whose |p-q| is 0. If we take the magnitude of  |p-q| into consideration, then the similarity between the two identical samples is then 0, which is clearly a wrong conclusion.
>
> 2. Further visualisation of lpNTK
>
> > Although Figure 2 and the displayed evolution of distinct FPC clusters is insightful, I was wondering whether additional visualisation of the lpNTK would have been possible?
>
> Thanks for pointing out the insights from the results. We were wondering if the reviewer has particular quantity to track during the FPC clustering? Since the computation complexity of FPC clustering is $\mathcal{O}(N log M)$, doing it every epoch is quite expensive, so we only did it after the convergence of the model.
>
> We are happy to provide further visualisation if there’s any interesting quantity in mind to track and visualise.
>
> 3. Results in Table 1
>
>  > The results in Table 1 showing the forgetting events don't seem particularly impressive (especially on MNIST). Am I missing something here?
>
>  We thank the reviewer for bringing this point up. Indeed, the prediction of forgetting event is not so impressive. We did the prediction with eNTK before, and it’s quite accurate (>90%). When predicting the forgetting events with lpNTK, we found there's a dilemma. If we do this after the convergence of the model, the lpNTK is more accurate but the forgetting events rarely happened. On the other hand, like in the paper, if we do the prediction at the beginning of the training, there are enough many forgetting events but the lpNTK is relatively inaccurate. However, in whichever case, the prediction performance is much better than random guessing, which can show that lpNTK helps a lot on the prediction of forgetting events.
>
> 4. Visualisation of the pruned samples.
>
> > It would have been great if the authors had provided a visualization of some of the pruned samples. Are there any patterns present in pruned examples in particular?
>
> We thank the reviewer for bringing up this interesting question. In fact, the samples in Figure 9 and Figure 10 are very likely to be pruned, as they are the most interchangeable ones to the centroid sample of the head cluster. Qualitatively speaking, one pattern we observed is that they are very typical and their feature is shared across each other. It will be very interesting to explore whether it is possible to quantify our observation.
>
> 5. Formatting issue
>
> > It is evident that the authors tried really hard to fit the paper into the page limit. There are formatting artifacts in terms of very little vspace throughout the paper, especially towards the end of the paper (pages 7-9). I would strongly encourage the authors to reduce dangling words instead of resorting to these very evident formatting tricks.
>
> We thank the reviewer for the suggestion, we've removed dangling words, and made some space for the last few sections.

---

> ### Author Response · Authors · 2023-11-19
> **Reply to reviewer 9biR (2/2)**
>
> 6. Connections between interchangeable/contradictory samples and forging/machine unlearning
>
> > Have the authors considered drawing connections between interchangeable and contradictory samples and forging / machine unlearning [1], i.e. by training on a specific point we could plausibly say that we have also optimized for another datapoint?
>
> We thank the reviewer for giving this interesting reference. For the interchangeable $x_1$ and $x_2$, we observe that training on $x_1$ will enhance the model’s confidence of $x_2$. Hence, it is possible that “optimising one also optimised another”. Actually, that is what a model does for a data sample in the test set (our model learns from $x_1$ in the training set, and predicts another $x_2$ in the test set well). For the unlearning and contradictory samples (say $x_1$ and $x_3$), we believe the phenomenon suggested in [1] might happen if some conditions are satisfied. For example, if the task is binary classification and $x_1$ has a different label than $x_3$ (but they are indeed quite similar). The update of $(x_1, y = +)$ will make the model become less confident about $(x3, y = -)$, which can be considered as “unlearning”. But when $K$ is large, things would become much more compicated. We will explore this interesting question in our future work about introducing the label modification into lpNTK.

---

> > ### Comment · Reviewer_9biR · 2023-11-23
> > **Thank you**
> >
> > I thank the authors for their rebuttal which has addressed many of my concerns. Although I still believe that more visualizations for the pruning behavior would have been promising, I consider the contribution of the paper as well as the author's responses to my concerns and their update paper draft enough to increase my score.

---

### Official Review · Reviewer_4TQg · 2023-11-11

**Soundness:** 2 fair
**Presentation:** 2 fair
**Contribution:** 2 fair
**Rating:** 6
**Confidence:** 4

**Summary:**

This paper enhances the Neural Tangent Kernel by integrating label information, which offers a more nuanced understanding of the interaction between samples compared to the existing eNTK method. The authors explore the relationship between lpNTK and eNTK, and use vector angles to classify samples into interchangeable, unrelated, and contradictory categories. This novel categorization facilitates a 'data-centric' improvement in model training. The idea is interesting, but there lack baselines to validate their proposed method.

**Strengths:**

1. The integration of label information into the Neural Tangent Kernel (NTK) represents a novel approach that enhances the characterization of sample interactions during optimization. And the sign approximation also makes sense.
2. The application of the proposed lpNTK is both reasonable and beneficial. It effectively validates the utility of lpNTK's mapping functions (vectors), demonstrating their practical effectiveness.

**Weaknesses:**

I have the following concerns:
1. Concerning Theorem 1, the authors assert that 'the gap between lpNTK and eNTK will not be significantly different.' However, this seems contradictory to subsequent analysis and empirical studies presented. The paper's central theme appears to be the integration of label information for a more nuanced understanding of sample relationships. If the lpNTK kernel closely resembles the original eNTK, could the authors clarify how this supports the stated claim? This warrants further explanation.
2. The categorization of samples into three groups mirrors the 'data-IQ' framework in Seedat et al. (2022), which also segments samples into easy, ambiguous, and hard categories. Data-IQ assesses prediction error and confidence throughout the training process, a concept seemingly echoed in this paper. I recommend that the authors draw comparisons with this methodology to highlight distinct aspects and contributions of their work.
3. There lack baselines to validate the effectiveness of the proposed method for measuring sample similarity. For instance, the influence function is a known technique for understanding relationships between samples during training. A comparison with such established methods would provide a more robust validation of the proposed approach.
4. It would be beneficial if the authors could include a computational complexity analysis of the lpNTK. This information would be crucial for understanding the practicality and scalability of the proposed method in different settings.

[1] Seedat, N., Crabbé, J., Bica, I., & van der Schaar, M. (2022). Data-IQ: Characterizing subgroups with heterogeneous outcomes in tabular data. Advances in Neural Information Processing Systems, 35, 23660-23674.

**Questions:**

Please refer to Weaknesses.

---

> ### Author Response · Authors · 2023-11-19
> **Reply to reviewer 4TQg (1/2)**
>
> We thank the reviewer for identifying the novelty of this work, as well as the practical potential of our lpNTK. Regarding the weaknesses, our responses are given as follows.
>
> 1. Inconsistent expression following Theorem 1
>
> > Concerning Theorem 1, the authors assert that 'the gap between lpNTK and eNTK will not be significantly different.' However, this seems contradictory to subsequent analysis and empirical studies presented. The paper's central theme appears to be the integration of label information for a more nuanced understanding of sample relationships. If the lpNTK kernel closely resembles the original eNTK, could the authors clarify how this supports the stated claim? This warrants further explanation.
>
> We thank the reviewer for pointing out this inconsistency, which might harm the understanding of the whole story of the paper. We've updated the Theorem 1 to avoid the following misconception: readers might think when the width ($w$) increases, the RHS of equation 4 will converge to 0, and hence lpNTK converges to eNTK in terms of F-norm. However, the original equation 4 says |lpNTK - eNTK| / |eNTK| converges to $O(w^{-1/2})$. If we move the |eNTK| term, which is $O(w^{1/2})$, to RHS, the new Theorem 1 becomes |lpNTK - eNTK| → O(1). That is to say, even under the infinity width assumption, the gap between lpNTK and eNTK is relatively bounded (not diverging to an arbitrarily large value) and non-negligible (not converging to 0). We also added Appendix B.2 to interpret more about the Theorem 1.
>
> 2. Categorisation of easy, ambiguous, and hard samples
>
> > The categorization of samples into three groups mirrors the 'data-IQ' framework in Seedat et al. (2022), which also segments samples into easy, ambiguous, and hard categories. Data-IQ assesses prediction error and confidence throughout the training process, a concept seemingly echoed in this paper. I recommend that the authors draw comparisons with this methodology to highlight distinct aspects and contributions of their work.
>
> We thank the reviewer for providing this good reference. We will add some discussion between our method and Data-IQ in the next version of the paper. In short, there are two major differences between these two works. First, they consider the difficulty of samples from different perspectives. Data-IQ focuses more on the properties of individual samples, while lpNTK focuses more on the interaction between different samples. Second, Data-IQ evaluates the difficulty by observing the learning curve, while lpNTK can explain why such curves might emerge. We believe the findings of these two works can support each other (e.g., contradictions in lpNTK might be a sign of the existence of a hard sample, which could be a high-confidence prediction with a wrong label.)
>
> 3. Comparison with influence function
>
> > There lack baselines to validate the effectiveness of the proposed method for measuring sample similarity. For instance, the influence function is a known technique for understanding relationships between samples during training. A comparison with such established methods would provide a more robust validation of the proposed approach.
>
> We thank the reviewer for pointing out the influence function as a possible baseline. We will add influence-score as one of the baselines for the experiments in Section 4.3 in the next version. In the meantime, we also mention that, influence-score is usually not as good as EL2N or forgetting score on CIFAR datasets [1]. So, the validation in Section 4.3 is very likely to be unaffected. In fact, in our extension work of lpNTK, our lpNTK-inspired data pruning method outperforms [1] and other existing baselines, which can help to show the practical effectiveness of lpNTK.
>
> Moreover, influence function focuses on how one sample might change the estimated parameters thus the predictions on other samples, whereas lpNTK focuses on the relationship between samples in the lpNTK representation space (a transformed gradient space).
>
> [1] Yang, S., Xie, Z., Peng, H., Xu, M., Sun, M., & Li, P. (2022). Dataset pruning: Reducing training data by examining generalization influence. ICLR-2023.
>
> [2] Koh, P. W., & Liang, P. (2017, July). Understanding black-box predictions via influence functions. In *International conference on machine learning* (pp. 1885-1894). PMLR.
>
> (Continued below)

---

> > ### Author Response · Authors · 2023-11-22
> > **Further updates on the comparison with influence function**
> >
> > We've added the performance of influence-score into the up-to-date Table 3. In short, influence-score performs slightly worse than all other baselines on CIFAR-10, while it performs roughly the same to the forgetting score on MNIST.
> >
> > Our new results solve the reviewer's concern about the comparison between our method and the influence function.

---

> > > ### Comment · Reviewer_4TQg · 2023-11-23
> > > **Thank you**
> > >
> > > Thank you for the further explanations, I hope the authors could add more discussions in their next version. Based on the rebuttal, I would like to raise my score to 6.

---

> ### Author Response · Authors · 2023-11-19
> **Reply to reviewer 4TQg (2/2)**
>
> 4. Discussion about the computational complexity of lpNTK
>
> > It would be beneficial if the authors could include a computational complexity analysis of the lpNTK. This information would be crucial for understanding the practicality and scalability of the proposed method in different settings.
>
> We thank the reviewer for pointing out this important problem. We've updated the discussion to cover the computational complexity of lpNTK. In short, the computational complexity of lpNTK is $\mathcal{O}(N^2d^2)$ where $N$ is the number of samples and $d$ is the number of parameters.
>
> The bottleneck of calculating lpNTK is mainly the size of the dataset: larger $N$ means a larger matrix (the influence of the number of classes K is circumvented by using the pNTK trick). We are considering to reduce the cost of this part using proxy quantities. For example, the learning-curve-based method (e.g., C-score or Data-IQ), or quantities like losses. Once the easy sample is identified, it is then not necessary to calculate $K(x_1, _x2)$ if $x_1$ and $x_2$ are both easy samples and look similar to each other.
>
> Furthermore, we can also use our method in some domains where the dataset size is not that big, e.g., alignment for LLM (roughly 10k samples needed) or few-shot learning. We would leave these to our future work.

---

### Official Review · Reviewer_3vgc · 2023-11-15

**Soundness:** 3 good
**Presentation:** 2 fair
**Contribution:** 3 good
**Rating:** 6
**Confidence:** 2

**Summary:**

This paper proposed to incorporate label information into the Neural Tangent Kernel (NTK) and designed a kernel called lpNTK to study the interaction between training examples. The author suggested classifying the relationships between a pair of examples into three types --- interchangeable, unrelated, and contradictory --- based on the angles between the vectors represented in lpNTK. The author then used these concepts to analyze some phenomena and techniques of learning dynamics, such as learning difficulty, forgetting, pruning, and redundancy. The observations and analyses were supported by experiments on the MNIST and CIFAR10 datasets.

**Strengths:**

Disclaimer: I'm not very familiar with the literature and some technical details of NTK. I might be biased because other reviews are visible to me before I write mine.

- The learning dynamics and learning difficulty is an important problem in many subfields of machine learning. The author provided nice theoretical tools for it based on NTK.
- Incorporating label information is useful for analyzing many supervised learning tasks in a more fine-grained way. The use case of data pruning is reasonable and convincing. I believe that the proposed tools can be used to deepen our understanding of some methods for learning from noisy/imbalanced data.
- This paper is well structured. The author raised intuitive hypotheses, asked clear questions, and then conducted reasonable experiments to verify them.
- The author contextualized this paper well and discussed related work sufficiently.

**Weaknesses:**

- Maybe it's because I'm unfamiliar with the literature, but I feel that this paper can benefit from mathematically clearer definitions of some terms such as interaction and learning difficulty.
- The author stated that "contradictory samples are rare in practice" but didn't explain why. I suspect that it's because the MNIST and CIFAR10 datasets used in experiments are relatively clean, and there are few ambiguous training examples. The conjectures in C.3 were nice, but I would expect more solid explanations or explorations.
- The author did not discuss much about the limitations of this work.

Minor issues:
- Section 2.1: the abbreviation $\mathbf{z} \in \mathbb{R}^K$ is misleading because I think $\mathbf{z}$ is a $\mathbb{R}^K$-valued function, not just a vector.
- Rigorously, a simplex with $K$ values is $(K-1)$-dimensional, i.e., it should be $\Delta^{K-1}$.
- Since many methods for noisy label learning and selective prediction (classification with rejection) heuristically make use of the learning dynamics, it would be convincing to apply lpNTK to those applications. However, those can be future work directions.

**Questions:**

- It is not completely clear to me why it is reasonable to fix the similarity matrix $\mathbf{K}$. Isn't it that a pair of training examples can be similar or dissimilar during different stages of training? How can we obtain the model that performs the best on the validation set in applications like data pruning?

---

> ### Author Response · Authors · 2023-11-19
> **Reply to Reviewer 3vgc (1/2)**
>
> We thank the reviewer for identifying the contribution of our work to the fundamental machine learning questions. We respond to the reviewer’s concerns one-by-one below.
>
> 1. Clearer definitions of terms
>
>  > I feel that this paper can benefit from mathematically clearer definitions of some terms such as interaction and learning difficulty.
>
> We thank the reviewer for the suggestion. We fixed the minor problems pointed out in the minor issues above to make the definitions of $\mathbf{z}$ and $\mathbf{q}$ more rigorous.  Regarding the definition of learning difficulty, in this paper, we measure it by accumulated loss over training, and interpret it as a result of pair-wise influence between samples. We agree that mathematically clearer definitions can further improve the formulation of the problem. Regarding the interaction between samples, in short, given a threshold $e$, intuitively, we can have the following relationships:
>
> - interchangeable: $lpNTK(x_1, x_2) \gg e$
>
> - unrelated: $-e < lpNTK(x_1, x_2) < e$
>
> - contradictory: $lpNTK(x_1, x_2) \ll  e$
>
> However, as this relative difficulty is a qualitative measurement, which highly depends on the dataset and network, we find it hard to define it rigorously.
>
> 2. Rare contradictory samples in practice
>
> > The author stated that "contradictory samples are rare in practice" but didn't explain why. I suspect that it's because the MNIST and CIFAR10 datasets used in experiments are relatively clean, and there are few ambiguous training examples. The conjectures in C.3 were nice, but I would expect more solid explanations or explorations.
>
> Thanks a lot for this good question. We also believe that the labels in benchmarks like MNIST and CIFAR are relatively clean, and hence contradictory samples are rare in practice. Actually, we believe that if the input signal is **sampled uniformly**, then the more complex the input samples are (e.g., colour images are more complex than black-and-write ones, high-resolution images are more complex than low-resolution ones) the less likely that the contradictory samples occur. However, in practice, there are indeed specific cases that can introduce many contradictory signals. For example, manually flipping the label, systematical change of the data collector, or for the object we are interested in is quite nuance in the image (like the two images with exactly the same background, but the people there are different). We added Appendix C.4 to specifically discussion the rare case of contradictory samples in practice.
>
> 3. No enough discussion about the limitation of this work
>
> > The author did not discuss much about the limitations of this work.
>
> Thanks for pointing out this. We have updated the discussion subsection of Section 4 to focus more on the practical limitations of this work. In short, the main limitation of this method is the computation of lpNTK, which highly depends on the number of input samples. Moreover, the computational complexity is $\mathcal{O}(N^2d^2)$ where $N$ is the number of samples and $d$ is the number of parameters. Also, if the model’s behaviour cannot be well approximated by NTK theory, the method proposed here will also be influenced.
>
> (Continued below)

---

> ### Author Response · Authors · 2023-11-19
> **Reply to Reviewer 3vgc (2/2)**
>
> 4. Fixed similarity matrix K
>
> > It is not completely clear to me why it is reasonable to fix the similarity matrix $K$. Isn't it that a pair of training examples can be similar or dissimilar during different stages of training? How can we obtain the model that performs the best on the validation set in applications like data pruning?
>
> We thank a lot for the reviewer to point this out. In fact, we’re working on a data-pruning method based on lpNTK. The limitation of lpNTK in the dynamic case is its computational complexity. So, to make it possible to approximate the lpNTK between samples during the training, we find a relatively good enough and practical proxy, i.e. the loss of samples. Following our analysis in Section 4.3, easy samples contribute not much to the generalisation performance. So, in our data pruning work based on lpNTK, we can prune samples that have similar lpNTK representations, thus smaller losses, without significantly decreasing the generalisation performance. Our experiments on ImageNet-1K shows that this lpNTK-based pruning method indeed outperforms the existing SOTA data pruning methods.
>
> Regarding the stability of $K$, it is true that the similarity matrix $K$ would behave quite differently during training. Specifically, at the beginning of pre-training, $K$ changes a lot and we cannot get too much useful information in this phase. On the other hand, at the very end of the training, $K$ jitters around a stable value, because most of the samples are already learned well, which is also not what we want. We speculate that is because the NN does feature learning at the beginning of training and then does the NTK-style learning after training for several epochs. In our experiments, we chose the model in this phase to calculate lpNTK, i.e., trained but not overfitted yet (we can observe the validation curve to select it, the phase change is quite obvious and the value of $K$ is quite stable in this phase). We added an outline for computing lpNTK in practice at the end of Section 2.3.
>
> 5. Noisy label issue
>
> > Since many methods for noisy label learning and selective prediction (classification with rejection) heuristically make use of the learning dynamics, it would be convincing to apply lpNTK to those applications. However, those can be future work directions.
>
> We appreciate the reviewer for pointing out this interesting direction. This is indeed very intriguing and promising. We will explore that in our future work.

---

### Comment · Area_Chair_Hcjw · 2023-11-10
**Authors-Reviewers discussion starts today, ends on Nov 22**

Dear authors and reviewers,

@Authors: please make sure you make the most of this phase, as you have the opportunity to clarify any misunderstanding from reviewers on your work. Please write rebuttals to reviews where appropriate, and the earlier the better as the current phase ends on Nov 22, so you might want to leave a few days to reviewers to acknowledge your rebuttal. After this date, you will no longer be able to engage with reviewers. I will lead a discussion with reviewers to reach a consensus decision and make a recommendation for your submission.
IMPORTANT: your paper is lacking one or more reviews -- we are working very hard to solve this by contacting reliable emergency reviewers. Please check this page daily as new reviews are likely to appear.

@Reviewers: please make sure you read other reviews, and the authors' rebuttals when they write one. Please update your reviews where appropriate, and explain so to authors if you decide to change your score (positively or negatively). Please do your best to engage with authors during this critical phase of the reviewing process.

This phase ends on November 22nd.

Your AC

---

### Author Response · Authors · 2023-11-19
**Overall Response**

We thank all the reviewers for recognising the novelty and contribution of this work, as well as the discussion about interesting directions for the future extension of our work.

Following the helpful suggestions and concerns from the reviewers, we made the following major updates to the work:

1. Updated Section 4.3 to discuss more about the limitations of the current work.

2. Updated the end of Section 2.3 to make it clearer about how to calculate lpNTK in practice.

3. Removed the original Section 4.1 to make spaces for the following section.

4. Added Appendix B.2 about the interpretation of Theorem 1.

5. Added Appendix C.4 to discuss why completely contradictory samples are rare in practice.

6. Added Influence Score as a baseline in Table 3.

We’re still updating the paper at the moment. To give reviewers enough time to read through our replies, and to carry out further discussion, we decided to post the updated version before the end of rebuttal. In some of our replies below, we said things to carry out in the future, and we are actually working on them.

We will upload another updated version of the paper in the next few days.

Thanks again for all the comments from the reviewers!

---

> ### Comment · Area_Chair_Hcjw · 2023-11-19
>
> Dear authors,
>
> Thank you for diligently submitting rebuttals. Please post a message here once you have submitted a revised version of your manuscript as otherwise reviewers (and myself) are likely to overlook it (we only get notifications for posts, not revised pdfs).
>
> AC

---

> ### Author Response · Authors · 2023-11-19
>
> Dear AC,
>
> Thanks very much for reminding us! We're updating the PDF at the moment. Once we've uploaded, we'll post a message to make sure you and all reviewers will be notified!
>
> Best,
>
> Submission 2038 Authors

---

> ### Author Response · Authors · 2023-11-19
> **Reminder about the updated PDF**
>
> Dear AC and reviewers,
>
> We write this message to remind you that we've uploaded an updated PDF following the suggestions and concerns from the reviewers.
>
> We're still working on polishing this work, and will update again in the next few days. We're also happy to answer any further questions that the reviewers may have.
>
> Thanks!
>
> Best,
>
> Submission 2038 Authors

---

### Author Response · Authors · 2023-11-21

Dear reviewers,

We are wondering if you have any follow-up questions about the updated paper, as well as our replies? We're happy to answer any further questions you may have.

Thanks!

Best,

Submission 2038 Authors

---

### Meta-Review · Area_Chair_Hcjw · 2023-12-05

**Metareview:**

This meta-review is a reflection of the reviews, rebuttals, discussions with reviewers and/or authors, and calibration with my senior area chair. This paper enhances the neural tangent kernel approach by integrating label information. This novel categorisation facilitates a 'data-centric' improvement in model training. The idea is interesting and should attract interest from the ICLR community, with promising results although the manuscript falls somewhat short in experiments as expressed by reviewers.

**Justification For Why Not Higher Score:**

Good contribution to a variation of the NTK, with promising results although falling somewhat short in experiments.

**Justification For Why Not Lower Score:**

Good contribution to a variation of the NTK, with promising results although falling somewhat short in experiments.

---

### Decision · Program_Chairs · 2024-01-16

Accept (poster)